# Into the Wild: A novel wild-derived inbred strain resource expands the genomic and phenotypic diversity of laboratory mouse models

Beth L. Dumont[1,2,3]*, Daniel M. Gatti[1], Mallory A. Ballinger[4], Dana Lin[5], Megan Phifer-Rixey[6], Michael J. Sheehan[7], Taichi A. Suzuki[8], Lydia K. Wooldridge[1], Hilda Opoku Frempong[1,3], Raman Akinyanju Lawal[1], Gary A. Churchill[1,2,3], Cathleen Lutz[1], Nadia Rosenthal[1,2,3,9], Jacqueline K. White[1], Michael W. Nachman[10]*

1 The Jackson Laboratory, 600 Main Street, Bar Harbor, Maine, United States of America, 2 Graduate School of Biomedical Sciences, Tufts University, Boston, Massachusetts, United States of America, 3 Graduate School of Biomedical Science and Engineering, The University of Maine, Orono, Maine, United States of America, 4 Department of Ecology and Evolutionary Biology, Cornell University, Ithaca, New York, United States of America, 5 Department of Biological Sciences, Vanderbilt University, Nashville, Tennessee, United States of America, 6 Department of Biology, Drexel University, Philadelphia, Pennsylvania, United States of America, 7 Department of Neurobiology and Behavior, Cornell University, Ithaca, New York, United States of America, 8 College of Health Solutions and Biodesign Center for Health Through Microbiomes, Arizona State University, Tempe, Arizona, United States of America, 9 National Heart and Lung Institute, Imperial College London, London, United Kingdom, 10 Department of Integrative Biology, Museum of Vertebrate Zoology, and Center for Computational Biology, University of California, Berkeley, Berkeley, California, United States of America

* beth.dumont@jax.org (BLD); mnachman@berkeley.edu (MWN)

## Abstract

The laboratory mouse has served as the premier animal model system for both basic and preclinical investigations for over a century. However, laboratory mice capture only a subset of the genetic variation found in wild mouse populations, ultimately limiting the potential of classical inbred strains to uncover phenotype-associated variants and pathways. Wild mouse populations are reservoirs of genetic diversity that could facilitate the discovery of new functional and disease-associated alleles, but the scarcity of commercially available, well-characterized wild mouse strains limits their broader adoption in biomedical research. To overcome this barrier, we have recently developed, sequenced, and phenotyped a set of 11 inbred strains derived from wild-caught *Mus musculus domesticus*. Each of these "Nachman strains" immortalizes a unique wild haplotype sampled from one of five environmentally distinct locations across North and South America. Whole genome sequence analysis reveals that each strain carries between 4.73–6.54 million single nucleotide differences relative to the GRCm39 mouse reference, with 42.5% of variants in the Nachman strain genomes absent from current classical inbred mouse strain panels. We phenotyped the Nachman strains on a customized pipeline to assess the scope of disease-relevant neurobehavioral, biochemical, physiological, metabolic, and morphological trait variation. The Nachman strains exhibit significant inter-strain variation in >90% of 1119 surveyed traits and expand the range of phenotypic diversity captured in classical inbred strain panels. These

---

**Data Availability Statement:** Whole genome sequences generated as part of this project are available on the NCBI Short Read Archive under

project accession number PRJNA1037121. SNP and structural variant calls are available for download at: https://doi.org/10.5281/zenodo. 10728008. All additional data, including raw and processed phenotype data, are provided in the Supporting Information files.

**Funding:** Importation, whole-genome sequencing, and phenotyping were supported by a JAX Director's Innovation Fund Award to B.L.D., G.C., N.R., C.L, and J.K.W. Genome sequencing of the two Tucson strains was partially supported by JAX Scholar Funds to R.A.L. B.L.D. is supported by a Maximizing Investigators' Research Award from the National Institute of General Medical Sciences (GM133415; https://www.nigms.nih.gov/). MWN was supported by GM074245, GM127468, and GM149304 from the National Institute of General Medical Sciences (https://www.nigms.nih.gov/). The funders had no role in study design, data collection and analysis, decision to publish, or preparation of the manuscript.

**Competing interests:** The authors have declared that no competing interests exist.

novel wild-derived inbred mouse strain resources are set to empower new discoveries in both basic and preclinical research.

## Author summary

Inbred laboratory mouse strains are integral tools for both preclinical and basic research. However, laboratory mice capture a small fraction of the genetic diversity found in wild mouse populations, a consideration that necessarily constrains the scope of possible discovery in studies restricted to lab mice. Further, laboratory mice were developed through programs of intense artificial selection for increased breeding performance, docility, and other traits of interest. Consequently, the genetic control of many complex traits in lab mice may not accurately reflect the mechanisms of their regulation in nature. To overcome these limitations, we developed a new inbred mouse strain resource founded from wild-caught mice subject to minimal laboratory selection. We show that strains in this "Nachman panel" harbor millions of genetic variants absent from current laboratory mouse models, including predicted deleterious alleles and gene-spanning structural variants. Paralleling this genetic diversity, we show that Nachman strains capture striking phenotypic variation across a multitude of disease-relevant biochemical, neurobehavioral, physiological, morphological, and metabolic traits, expanding the range of trait variation recovered in lab strains alone. Overall, our strain survey emphasizes the collective potential of the Nachman strains to advance discoveries into multiple disease areas and basic biology.

## Introduction

Inbred mouse strains have served as the workhorses of mammalian genetics for over a century [1]. Standardized inbred strain backgrounds ensure experimental reproducibility across labs and experiments, provide the backbone for mechanistic investigations into gene and pathway function, and enable testing of genetically identical cohorts across different treatments, exposures, and perturbations. Inbred strains also provide platforms for community resource development, including comprehensive gene knockout panels [2, 3] and phenome resources [4].

The classical inbred (CI) mouse strains were developed from a small number of founder mice purpose-bred by mouse fanciers for traits of interest in the early 1900s [5,6]. As a result, the CI strains capture a limited subset of the genetic variation found in wild mouse populations [7,8]. Many CI strains have inherited large stretches of their genome identical by descent, such that pairwise strain comparisons yield numerous genomic regions where segregating variation is reduced to or near zero [8,9]. Furthermore, due to their unique origins and history of selective breeding, the complex architecture of trait variation in inbred strains may not faithfully model the genetic architecture of phenotypic variation in human populations [10].

Wild mouse populations harbor many predicted functional and disease-associated alleles, the majority of which are not present in CI mouse strains and have therefore never been experimentally tested in the laboratory [8,11]. Thus, wild mice present an untapped opportunity for advancing new biomedical research discoveries [7,8,11]. In contrast to the history of intense artificial selection and subspecies admixture that has molded the genomes of classical laboratory strains, the genetic diversity observed in wild mice reflects the interplay of selection, genetic drift, mutation, and migration, mirroring the natural population genetic processes that

have sculpted the contemporary landscape of human genomic diversity. Wild mice may therefore better approximate the diversity and organization of functional genetic variation in human populations than CI strains, including genetic variation influencing responses to dietary challenges, pharmaceutical interventions, and toxin exposures.

Despite these potential advantages, multiple challenges stand in the way of using wild-caught mice in biomedical research. Trapping wild house mice is laborious, especially if one is interested in assembling a large sample of unrelated individuals. Further, wild mice are genetically unique, preventing experimental studies that require reproducible or controlled genetic backgrounds. In addition, phenotypic variation in wild mice is influenced by differences in age, reproductive history, health status, and other, typically unknown, environmental exposures. Finally, wild mice are also vectors for numerous pathogens that pose a threat to human health and the health status of laboratory mouse colonies.

Wild-derived inbred mouse strains (WDIS) present a powerful intermediary between CI and wild mice. WDIS are developed from wild-caught mice that are brother-sister mated in a laboratory environment for >20 generations, thereby immortalizing a single haplotype from the wild in an inbred state. Thus, WDIS combine the reproducibility and fixed genetic background of inbred mouse models with the natural diversity present in wild mouse populations. WDIS have been strategically utilized in gene mapping studies to introduce increased diversity [12–16], profiled in immunological [17], metabolic [18], and reproductive studies [19–21], and featured as founders to current mouse diversity populations such as the Collaborative Cross (CC) and Diversity Outbred (DO; [22,23]). Indeed, the majority of quantitative trait loci (QTL) identified in mapping studies in the CC and DO are attributable to the allelic effects of one or more of the three wild-derived founder haplotypes [24–27].

Despite their realized power, only a modest number of wild-derived inbred strains are commercially available (S1 Table). Of these, many are poor breeders, are maintained at low (or undocumented) health status, or have few associated genomic resources. These considerations present notable obstacles to their widespread use in biomedical research. Furthermore, whereas the genomes of CI strains are largely derived from one house mouse subspecies (*M. m. domesticus*) [6], many WDIS are representatives of alternative house mouse subspecies that exhibit variable degrees of reproductive isolation from *M. m. domesticus*. Crosses between these WDIS and CI strains often expose multilocus incompatibilities linked to hybrid sterility [28,29], an outcome that further limits their practical utility in genetic studies.

Recently, we developed a panel of ~25 WDIS from wild-caught *M. m. domesticus* from five locations across North and South America: Saratoga Springs, New York, USA; Gainesville, Florida, USA; Manaus, Brazil; Tucson, Arizona, USA; and Edmonton, Alberta, Canada (Fig 1). These sampling locations are defined by distinct ecosystems and climates, including tropical rainforest (Manaus), desert (Tucson), temperate forest (Saratoga Springs), prairie (Edmonton), and wetland (Gainesville) habitats. As a result, wild mice from these regions have been subject to distinct selective pressures and have evolved unique morphological and physiological adaptations to their environments [30–33]. Recent work has also uncovered dramatic transcriptomic differences between populations, revealing divergence at the functional genomic level [32]. Taken together, data from wild mice sampled from these five geographic regions portend the rich potential for their descendant inbred strains to serve as powerful new mouse models of susceptibility and resilience to numerous diseases, phenotypes, and conditions relevant to human health.

From 2019–2022, we imported a representative subset of 11 WDIS from the broader parent Nachman panel to The Jackson Laboratory (JAX), including at least one inbred line per geographic location (S2 Table). Here, we introduce this novel diverse mouse strain resource, including the extent of genomic and phenotypic diversity across these strains and their

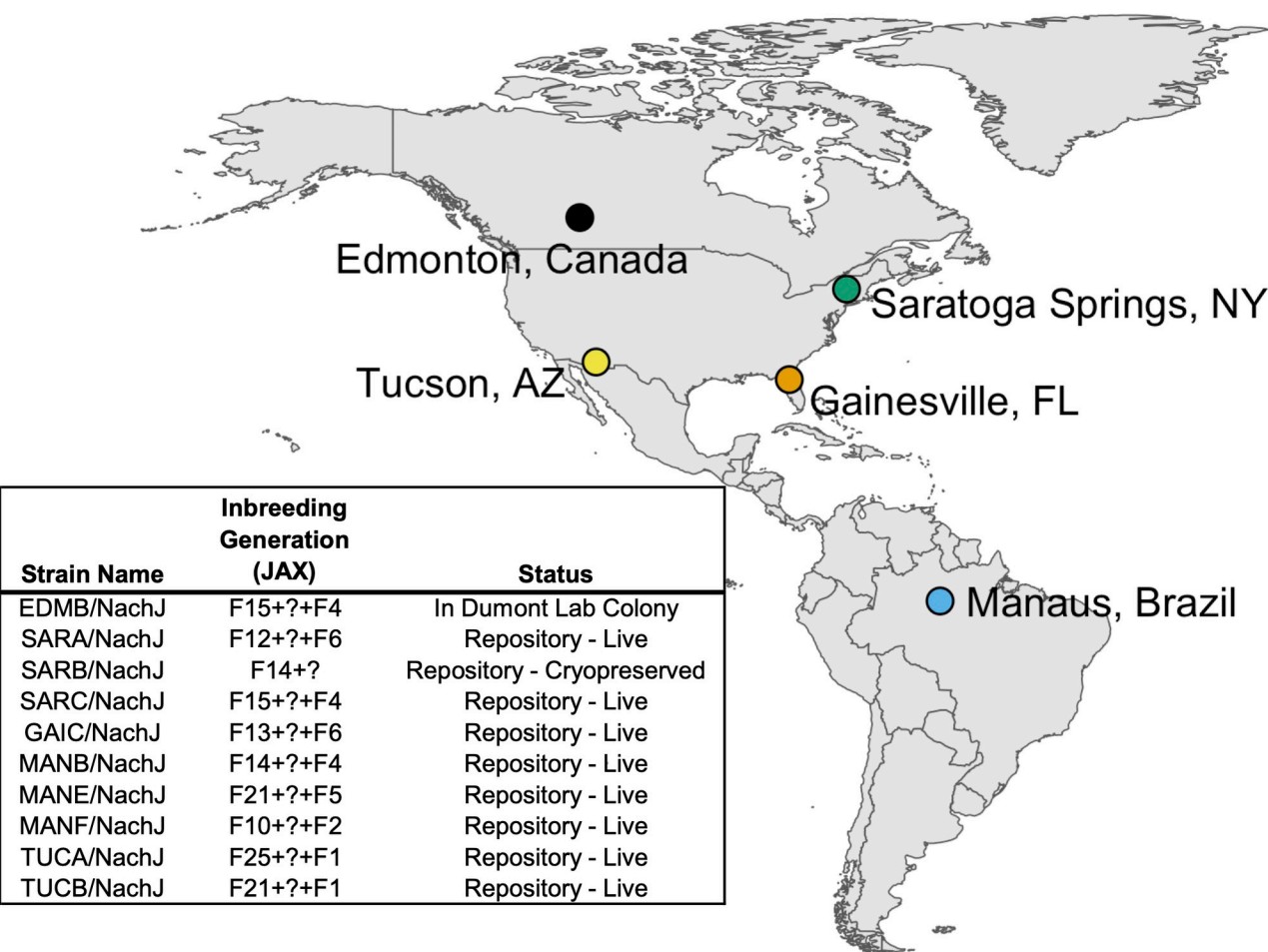

| Strain Name | Inbreeding Generation (JAX) | Status |
|---|---|---|
| EDMB/NachJ | F15+?+F4 | In Dumont Lab Colony |
| SARA/NachJ | F12+?+F6 | Repository - Live |
| SARB/NachJ | F14+? | Repository - Cryopreserved |
| SARC/NachJ | F15+?+F4 | Repository - Live |
| GAIC/NachJ | F13+?+F6 | Repository - Live |
| MANB/NachJ | F14+?+F4 | Repository - Live |
| MANE/NachJ | F21+?+F5 | Repository - Live |
| MANF/NachJ | F10+?+F2 | Repository - Live |
| TUCA/NachJ | F25+?+F1 | Repository - Live |
| TUCB/NachJ | F21+?+F1 | Repository - Live |

**Fig 1. Map of sample locations and summary of inbred strains imported to the JAX Repository.** Inbred strains were developed from wild-caught mice trapped at 5 sample locations across North and South America: Manaus, Brazil (MANB/NachJ, MANE/NachJ, MANF/NachJ); Saratoga Springs, New York, USA (SARA/NachJ, SARB/NachJ, SARC/NachJ); Gainesville, Florida, USA (GAIC/NachJ); Tucson, Arizona, USA (TUCA/NachJ, TUCB/NachJ); and Edmonton, Alberta, Canada (EDMB/NachJ). Strain GAIA/NachJ was successfully imported but has been since discontinued from the JAX Repository due to poor breeding and is not listed. Inbreeding generation numbers are presented in the following format: number of inbreeding generations in Nachman Colony prior to importation at JAX + number of inbreeding generations in JAX's Importation Facility + number of inbreeding generations in the JAX Repository. Inbreeding generation numbers are imprecise owing to the use of inter-generational crosses in JAX's importation facility to expedite colony expansion for oocyte harvests. This ambiguity is denoted by a "?" in the inbreeding generation supplied in the strain table. The map base layer was made with Natural Earth, which maintains all raster and vector map data in the public domain (https://www.naturalearthdata.com/about/terms-of-use/). Specifically, the base layer was produced using the *rnaturalearth* package for R by invoking the command ne_countries(continent = c("North America", "South America"), scale = "medium", returnclass = "sf").

relationship to CI strains. Our strain survey emphasizes the collective promise of these Nachman strains to advance biomedical discoveries into multiple trait domains, systems genetics analysis, and fundamental principles of evolutionary biology.

## Results

### Generating the Nachman Panel

Wild house mice were caught in 2012–2013 from five geographic locations (Fig 1). Animals were transported to UC Berkeley, and mice from each geographic region were randomly paired to create inbred lines which were propagated through brother-sister mating for at least 10 generations. Initially, ~10 independent lines were established from each location. No

attempts were made to rescue lines that exhibited infertility due to inbreeding depression and half of the initiated lines eventually became extinct. At the time of writing, 25 of the initiated lines remain in Dr. Nachman's strain holdings at UC Berkeley.

A subset of 20 strains were selected for importation to the JAX Repository (**S2 Table**). Of these strains, 11 were successfully rederived via *in vitro* fertilization (IVF) and embryo transfer to a pseudopregnant dam and integrated into production breeding colonies (**Fig 1**). Reasons for rederivation failure were complex and variable, ranging from poor breeding, failure to recover sufficient numbers of oocytes for IVF, no live births following multiple embryo transfers, and accidental strain contamination (**S2 Table**). One successfully imported strain (GAIA/NachJ) was subsequently terminated due to poor breeding performance.

## Breeding performance

Many wild-derived inbred strains breed poorly, a consideration that limits their practical utility in biomedical research. Our strategic propagation of only the best performing inbreeding lineages derived from each wild-caught founder pair should ensure that resulting inbred strains are vigorous breeders. Indeed, the inbred Nachman lines breed reliably across two independent animal facilities (**Fig 2**). Fewer than 25% of established matings are non-productive and average weaned litter sizes for most strains are between 3–6 pups.

While comprehensive diallele crosses have not yet been performed, there are no signs of F1 sterility or reduced fertility in the few F1 hybrids between independent Nachman strains that have been generated to date (**S3 Table**). To the contrary, testis weight and sperm density measurements in F1 hybrids exceed fertility metrics quantified in the inbred parental lines and classical inbred strains, revealing hybrid vigor (**S1 Fig** and **S3 Table**). Efforts to more comprehensively profile reproductive traits in the inbred Nachman strains and their derivative F1 hybrids are on-going, and we expect our future results to strengthen empirical support for the early trends reported here. We note that these preliminary findings fall in contrast to observations from diallele crosses between the inbred founder strains of the DO and CC [34], which include strains from reproductively isolated subspecies. Many incipient inbred CC strains were lost due to male infertility [35] and many surviving strains exhibit poor breeding performance, male infertility, and extreme sex ratio distortion [36]. Thus, strains in the Nachman WDIS panel comprise a shelf-stable mouse diversity resource with limited potential to uncover genetic incompatibilities in experimental crosses between Nachman strains or between Nachman strains and CI strains.

## Cytogenetic characterization of Nachman Strain Karyotypes

Wild *M. m. domesticus* populations harbor frequent Robertsonian chromosomal translocations that give rise to considerable karyotypic diversity across Europe [37]. Hybrid mice from crosses between different karyotypic races exhibit reduced fitness owing to altered chromosome dynamics and inefficient chromosome segregation during meiosis [37–40]. To confirm the absence of large-scale karyotypic alterations or changes in chromosome number among strains, we generated spermatoctye cell spreads from each inbred Nachman line and assessed karyotype and meiotic chromosome pairing by fluorescent microscopy (**S2 Fig**). These cytogenetic analyses are on-going, but all Nachman strains evaluated to date exhibit the standard 2n = 40 all-acrocentric karyotype (7 tested strains: EDMB, GAIC, SARA, SARB, MANB, MANF, and TUCB).

## Genomic diversity in the Nachman Panel

To assess the extent of genomic diversity in this new strain resource, we sequenced the whole genomes of a representative male from each strain to moderate coverage using PacBio HiFi

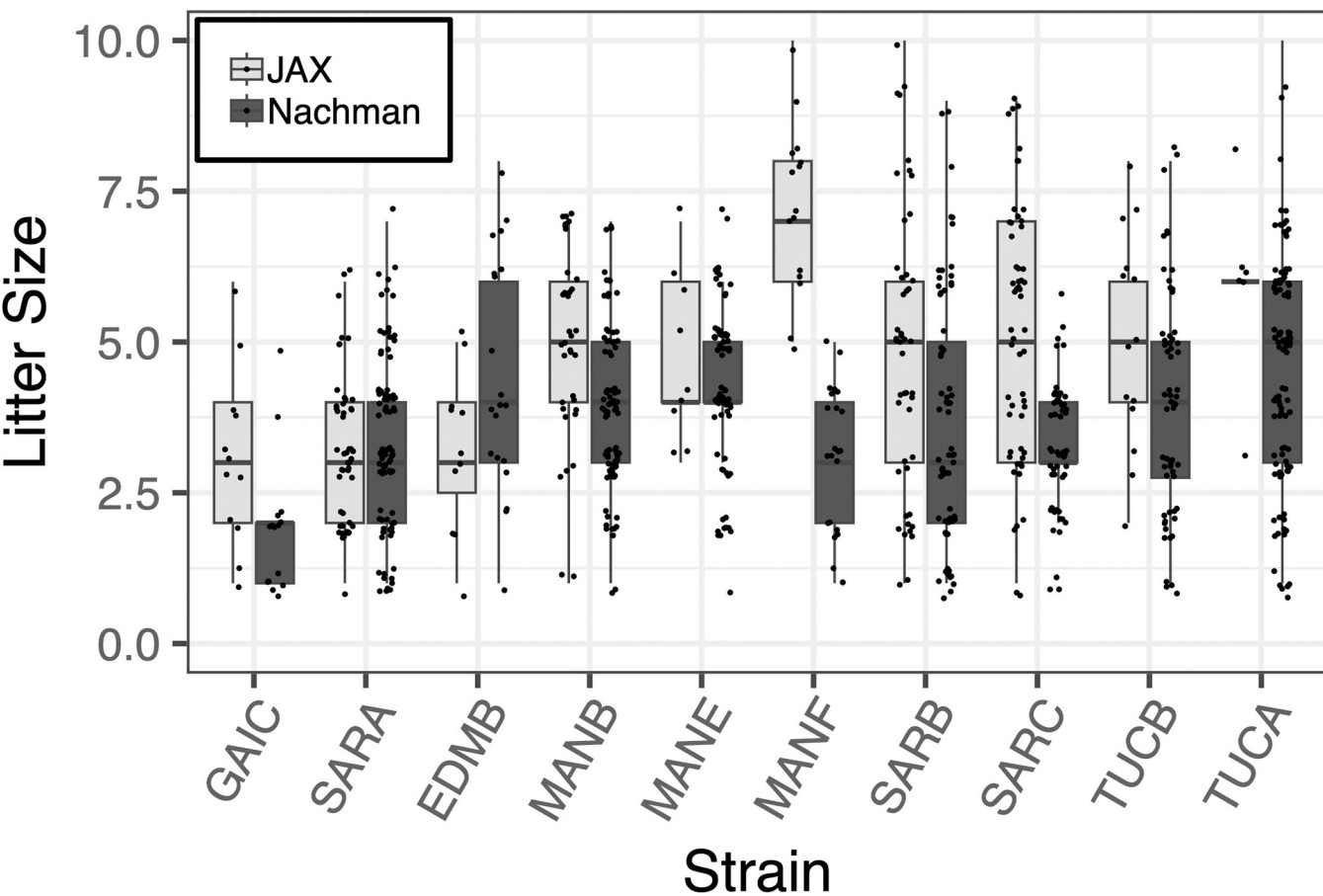

**Fig 2. Distribution of weaned litter sizes across 10 strains in the Nachman WDIS panel.** Sizes of litters born at JAX and in the Nachman Lab at UC Berkeley are presented as standard box plots, with box width defining the inter-quartile range and the thick line bisecting each box denoting the median. Litter sizes are jittered for ease of visualization.

sequencing technology (~10x coverage per strain; **S4 Table**). Average read lengths exceeded 10kb for all strains (range: 10.217–14.957kb), with >96.67% of sequenced bases exceeding a quality score of 30 (**S4 Table**). Sequenced reads were mapped to the GRCm39 reference genome assembly [41] and subject to single nucleotide variant (SNV) calling using DeepVariant [42, 43]. Each strain harbors between 4.73–6.54 million fixed SNV differences relative to the C57BL/6J-based reference, with >2.75M SNVs distinguishing any pair of Nachman strains (**Fig 3A**; corresponding to a minimum of ~1 SNP every ~1 kb). Although sequenced mice have undergone a modest number of inbreeding generations, within strain heterozygosity is low ($\pi < 0.0006$ versus ~0.0017 in wild-caught *M. m. domesticus*) and aligns with theoretical expectations for the number of inbreeding generations (**S3 Fig**).

We performed joint SNV calling with the Nachman lines and 51 CI mouse strains previously sequenced by the Mouse Genomes Project (44). Of the 16,071,877 autosomal SNVs observed in the Nachman strains, 6,836,536 variants (42.5%) are absent from CI strains [44]. We constructed a maximum likelihood phylogenetic tree from SNPs on chr19 to assess relationships between existing mouse strains and strains in this new wild-derived inbred strain resource. The CI strains present as a single clade nested within the diversity sampled in the Nachman panel (**Fig 3C**). Branch lengths are notably longer for the WDIS than for the CI strains, reflecting the greater diversity in the former. These findings are confirmed by a

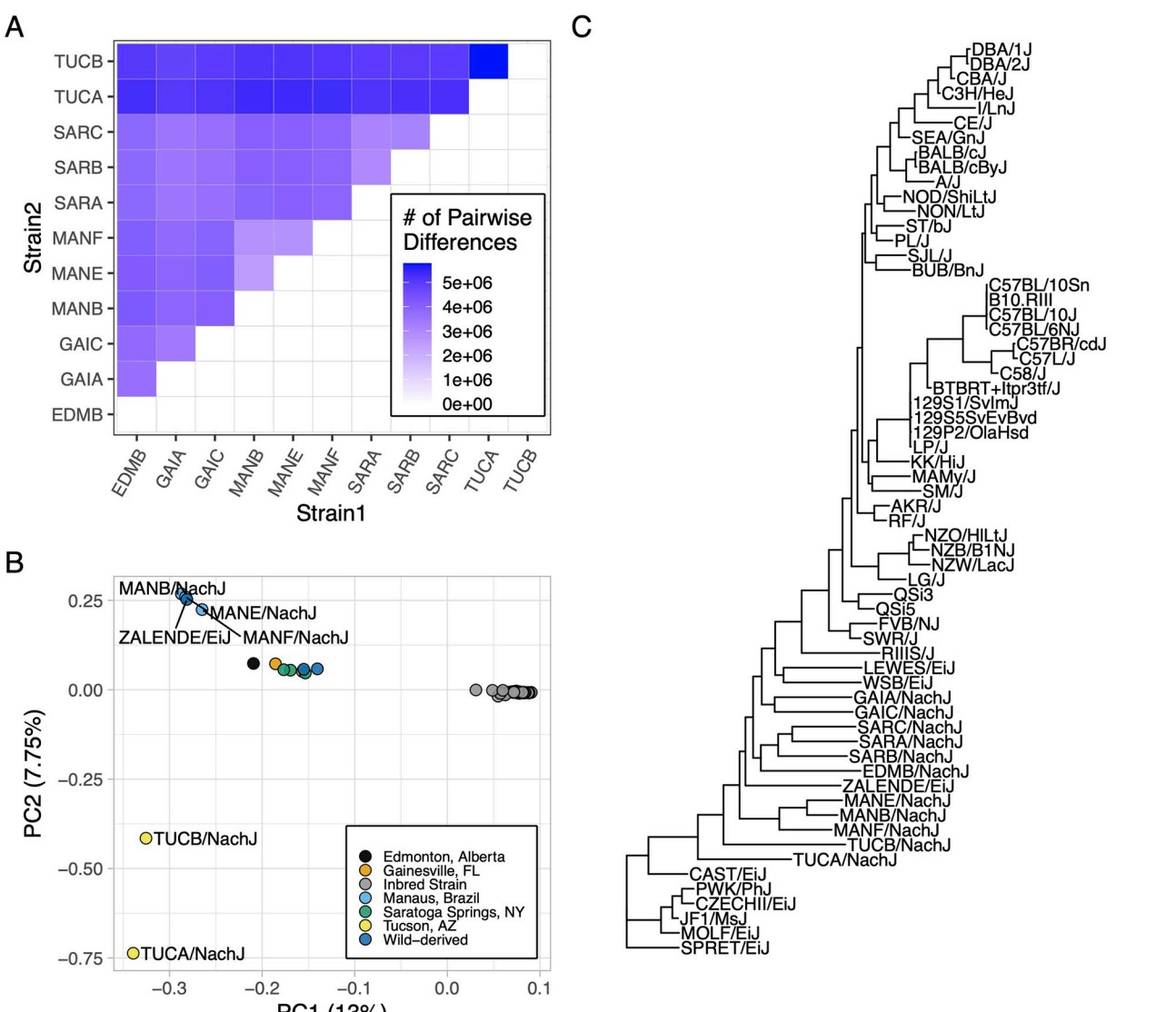

**Fig 3. Genetic diversity in the Nachman strains.** (A) Heatmap displaying the number of pairwise SNP differences between Nachman lines. Counts derive from the autosomal genome fraction only and exclude unplaced contigs. (B) PC analysis of autosomal genetic diversity partitions the Nachman lines by sample location and isolates Nachman strains from the CI mouse strains. (C) Maximum likelihood phylogenetic tree constructed from SNPs on chr19. The CI strains form a single clade nested within the diversity sampled by the Nachman strains.

principal component (PC) analysis, which reveals a single cluster of points corresponding to the CI strains (**Fig 3B**), with the Nachman strains and other WDIS well-separated in PC coordinate space. However, we acknowledge that differences in the technologies used to sequence the Nachman strains (PacBio HiFi) and other strains (Illumina) could partially contribute to the isolation of the Nachman strains along these PC axes. Nachman strains derived from a common locale are more genetically similar than strains from divergent locations, although strains from Gainesville, Edmonton, and Saratoga Springs are minimally separated along PC1-PC5 (41.53% of the variance). PC dimensions 6–8 provide separation across these geographic sample locations (**S4 Fig**).

We next annotated variants according to likely functional impact. Variant counts for different functional categories are provided in **S5 Table**. Overall, the Nachman strain panel harbors

1,976 SNPs predicted to be highly deleterious, with 1,319 of these variants not observed in classical inbred laboratory strains. These highly deleterious variants are enriched in genes with biological roles in sensory perception and G protein-coupled receptor signaling (**S5 Table**). The Nachman strains also harbor predicted loss-of-function alleles at genes implicated in human diseases. For example, the TUCA/NachJ and TUCB/NachJ lines harbor a premature stop variant in *Eif2b1* that is a predicted target of nonsense-mediated decay (NMD; chr5:124716942). *Eif2b1* encodes a subunit of the eukaryotic translation initiation factor eIF2B, which is essential for protein synthesis. Mutations in this gene have been linked to leukoencephalopathy with vanishing white matter and ovarian failure in humans [45,46]. Similarly, a predicted NMD variant in *Yeats2* (chr16:20028820, stop-gain mutation) is present in GAIC/NachJ, TUCA/NachJ, and TUCB/NachJ. Mutations in this gene are causally associated with myoclonic epilepsy in humans [47]. Strains MANB/NachJ, MANF/NachJ, TUCA/NachJ, and TUCB/NachJ carry a predicted loss-of-function splice-donor variant in *Ifnar1* (chr16:91302893). Humans with mutations in this gene have immunodeficiency-106 and exhibit hyperinflammatory responses to some vaccines [48].

Taken together, our analyses indicate that the Nachman strains harbor considerable genetic diversity that is not captured in existing inbred mouse strain panels. A subset of this variation is likely functional, establishing the prediction of broad phenotypic diversity among these strains and their phenotypic divergence from CI mouse strains.

## Structural diversity in the Nachman Panel

Structural variants (SVs) are important contributors to phenotypic diversity, including disease risk and incidence. Prior work has suggested that house mouse genomes are burdened by higher rates of structural mutation than human genomes [49,50], leading us to posit the presence of abundant, potentially functional SVs within the Nachman strain genomes.

Using an ensemble approach to minimize false positive SV calls (see Methods), we identified 274,177 autosomal structural variants across the 11 sequenced Nachman strain genomes (139,931 deletions, 133,058 insertions, 710 duplications, and 478 inversions). Of the 139,931 deletions discovered in the Nachman strains, 80,222 were not previously detected in a diverse panel of inbred strains (57.3%; requiring 75% reciprocal overlap [49]). Similarly, 60.6% of insertions ascertained in the Nachman panel are not present in the current mouse SV catalog [49] (n = 80,568 unique insertions in the Nachman lines). Nachman strain genomes each contain 13.5 Mb—18.3 Mb of sequence that is absent in the mm39 C57BL/6J-based reference genome (**Fig 4A**), with 23.5 Mb—30.6 Mb of sequence in the reference genome absent from any given Nachman strain (**Fig 4B**). The modest genome coverage of the Nachman lines may lead to an appreciable number of missed SVs (*i.e.*, false negatives) in these genomes, implying that SVs potentially have a greater collective impact on these genomes than suggested by the numbers presented here.

Many SVs in the Nachman strain genomes are potentially functional. Approximately 46.2% of SVs overlap RefSeq genes (126,741 SVs overlapping 18,504 unique genes), with 89,757 SVs overlapping the annotated coding regions of 12,209 unique genes. An additional 933 SVs lead to the ablation of whole transcripts (**S6 Table**). The majority of these transcript ablating SVs impact predicted genes or pseudogenes, although several olfactory receptors, vomeronasal receptors, and immunoglobulins harbor loss-of-function structural mutations.

Recent work has suggested that a majority of SVs in mouse genomes are due to transposable element (TE) activity [49]. We theorized that many SVs in the Nachman strains are likewise mediated by TE-activity in the genome. Indeed, the size distribution of insertion and deletion calls in the Nachman strain call set is consistent with key contributions from TEs (**S5 Fig**),

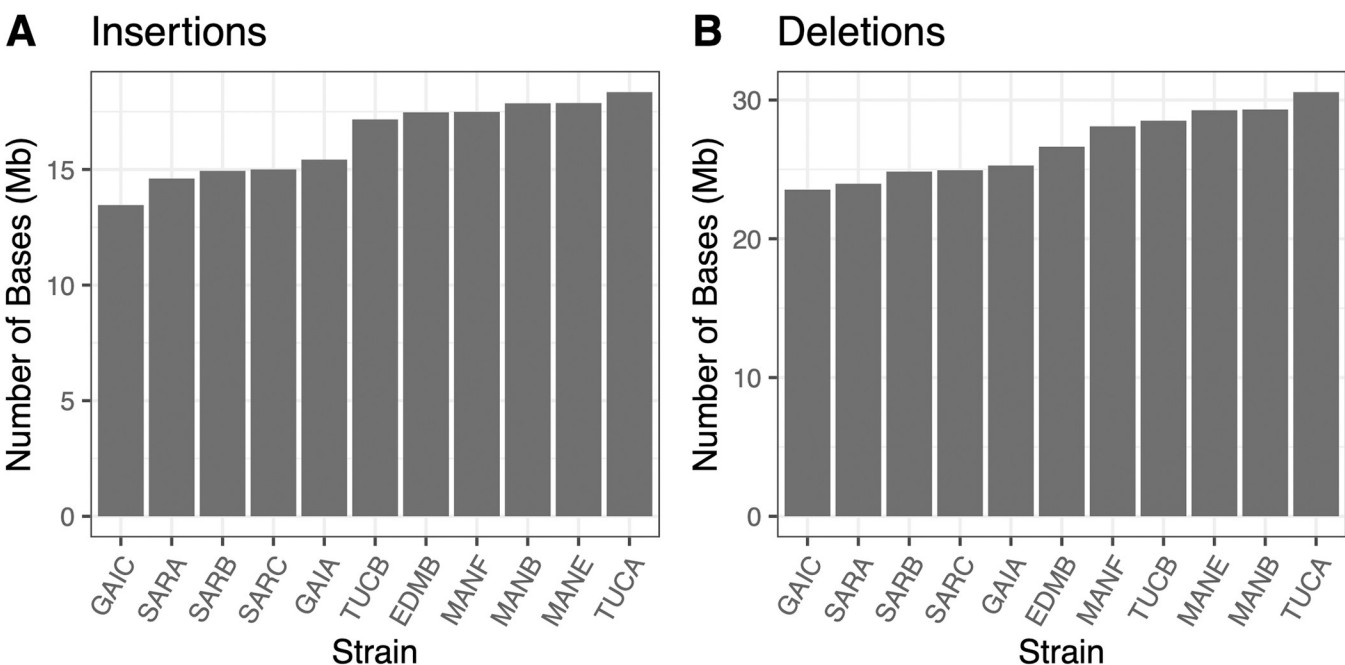

**Fig 4.** Numbers of bases impacted by (A) insertion and (B) deletion events in each Nachman strain relative to the mm39 reference sequence.

with peaks corresponding to the average length of SINEs and LINEs. To more formally assess this possibility, we annotated all insertion and deletion SV calls using repeatMasker (see Methods). Overall, we identify 113.7 Mb of SV-associated sequence in the Nachman strains comprised of TE-derived sequence (57.01Mb deletion, 56.70Mb insertion), corresponding to 78.1% of SV-impacted bases (145.76Mb). SINE B2 MM1A repeats are the most abundant TEs within SVs (15,169 polymorphic MM1A elements in the Nachman panel; **S7 Table**), consistent with prior reports of active SINE B2 transposition in house mouse genomes [49,51]. RLTR10, IAPEz, and L1MdGf elements are also highly abundant in SVs identified in the Nachman lines (**S7 Table**), consistent with the young age of these elements and their active mobilization in mouse genomes [49,51,52]. Overall, our results suggest that TEs have substantially contributed to the landscape of SVs in the inbred Nachman strains and, by extension, the wild mouse populations from which these strains derive.

## Strain subspecies ancestry

An earlier investigation of wild-caught mice from Tucson, Arizona uncovered evidence of introgression from *M. m. castaneus* [30]. We used a sample of 134 wild-caught mouse genomes from *M. m castaneus*, *M. m. domesticus*, and the outgroup *M. spretus* to evaluate genomic evidence of possible *M. m. castaneus* introgression in the TUCA/NachJ and TUCB/NachJ lines (**S8 Table**). Patterson's D is significantly non-zero for both strains ($D_{TUCA}$ = 0.213 and $D_{TUCB}$ = 0.192; P < 2.3 x $10^{-16}$ for both strains), with estimated *M. m. castaneus* admixture proportions ($f_4$ ratio) of 13.4% and 11.3%, respectively. *D* is also significantly greater than zero for SARB/NachJ and SARC/NachJ (*P* < 0.05; **S9 Table**). However, the estimated admixture proportion from *M. m. castaneus* is <1% in these strains and may be attributable to the incomplete sampling of ancestral wild mouse diversity. We conclude that the genomes of both Tucson lines, but not strains from other locations, harbor significant *M. m. castaneus* ancestry. We also investigated possible introgression from *M. m. musculus* and found that all Nachman

strains have <1% admixture from *M. m. musculus*, indicating no significant or recent introgression from this subspecies (**S9 Table**).

We next estimated admixture statistics in windows of 5000 informative SNPs (2500 slide) across the genomes of TUCA/NachJ and TUCB/NachJ, focusing on the 5% of windows with the highest $f_{dM}$ values to identify regions of likely *M. m. castaneus* introgression. We utilize $f_{dM}$, rather than the related *D* statistic, as the variance in *D* can be quite large when applied to small windows [53,54]. Introgressed regions are overwhelmingly unique to either TUCA/NachJ or TUCB/NachJ (**S10** and **S11** **Tables** and **S6 Fig**), consistent with the high genomic divergence between these lines (**Fig 3**). For strain TUCA/NachJ, genes in introgressed regions are enriched for biological processes related to regulation of lipopolysaccharide-mediated signaling, zymogen activation, and sensory perception of taste and smell (**S12 Table**). For strain TUCB/NachJ, genes in regions of *M. m. castaneus* introgression are enriched for biological processes related to testosterone biosynthesis, keratinization, and multiple aspects of immunity (**S13 Table**). Overall, introgressed regions are short, implying that events are not recent (**S6 Fig**).

## Relationship to wild mouse diversity

To contextualize the variation present in the inbred Nachman strains with that observed in wild *M. m. domesticus*, we created a joint variant callset featuring the 11 inbred Nachman strains and 97 publicly available wild *M. m. domesticus* samples from multiple populations (Iran, France, Germany, and the Eastern United States). PC analysis on this Nachman-wild mouse call set reveals genetic clustering of the Nachman lines with samples from Europe and the US, with samples from Iran isolated along PC1 (13.9% of the variance; **Fig 5A**). These trends are recapitulated by a maximum likelihood phylogenetic tree of these samples, which places the Iranian samples as ancestral to the Nachman strains and wild-caught mice from Europe and the United States (**Fig 5C**). Our findings are consistent with previous population genetic analyses of wild mice, which suggest that *M. m. domesticus* from the Indo-Iranian valley harbor elevated genetic diversity [11,55] and point to the European origin of North American house mice [56].

We excluded the Iranian samples and repeated the PCA to evaluate how the genetic diversity captured in the Nachman samples compares to the genetic variation in contemporary *M. m. domesticus* from Europe and the Eastern US (**Fig 5B**). PC1 (12.7% of the variance) stratifies mice from wild-caught *M. m. domesticus* populations, with Nachman lines falling at intermediate positions along this PC. PC2 (10.2% of the variance) isolates the two Nachman strains from Tucson, likely reflecting the presence of *M. m. castaneus* admixture in these lines. While the Nachman lines introduce significant new variation into laboratory strain collections, the 11 inbred strains in this panel sample only a subset of the genetic variation present in wild *M. m. domesticus* populations.

## Nachman wild-derived inbred strains capture extensive phenotypic diversity

We subjected mice from the majority of JAX imported Nachman lines to a 19-week phenotyping pipeline to profile strain variation in multiple metabolic, neurobehavioral, physiological, morphological and biochemical traits (**Fig 6**). Males and females from 9 strains were phenotyped across 16 cohorts of age-matched animals (+/-6 days). Cohorts were comprised of mice from multiple strains, and the majority of phenotyping cohorts included C57BL/6J control mice to permit post-hoc detection of potential batch effects. The strain composition of each cohort is provided in **S14 Table**. Overall, we collected 1119 phenotype measures from each

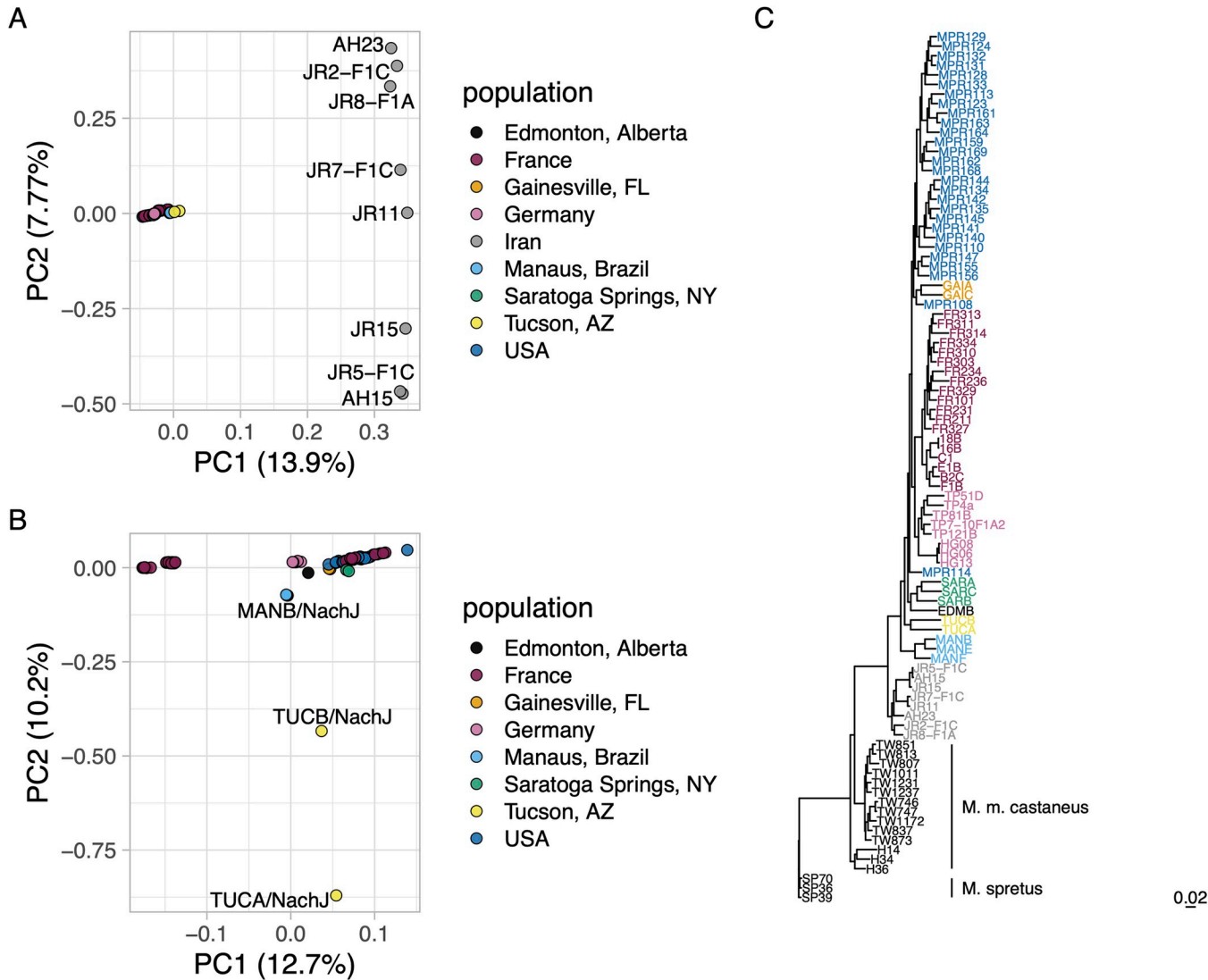

**Fig 5. Genetic diversity in Nachman strains and wild-caught *M. m. domesticus*.** (A) Principal component analysis performed on autosomal variants segregating in wild-caught *M. m. domesticus* mice from multiple populations and inbred Nachman strains. PC1 separates wild-caught mice from Iran and all other mice. PC2 stratifies mice from within Iran. (B) Excluding mice from Iran provides increased granularity to detect differences across other *M. m. domesticus* populations. PC2 isolates the Nachman strains from Tucson, Arizona from other strains and populations. (C) Maximum likelihood tree constructed from biallelic chr19 SNPs. Strains are color-coded according to the legends in A and B.

animal, although many trait values are highly correlated and therefore not independent (**S15 Table**).

More than 90% of the 1119 surveyed phenotypes differ among Nachman strains, with 86.7% and 80.4% of phenotypes differing significantly in comparisons involving only females or males, respectively (Kruskal-Wallis test, P<0.05; **S16 Table**). We obtain qualitatively identical results using both one-way ANOVA (**S17 Table**) and a linear modelling approach with strain treated as a random factor (73.8% of models including strain as a random effect variable provide significantly better model fit than a reduced model excluding strain; see Methods; **S18 and S19 Tables**). Approximately 25% (n = 288) of phenotypes exhibit significant differences between males and females and a significant strain-by-sex effect is observed for 21.5% (n = 235) of surveyed measures (two-way fixed effects ANOVA, $P < 0.05$; **S20 Table**). Similar

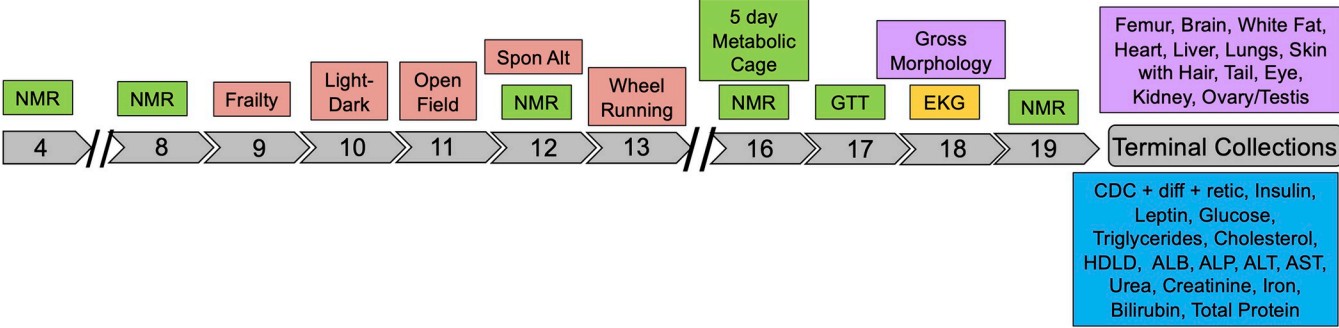

**Fig 6. Schematic of the phenotyping pipeline.** Mice were transferred to the Center for Biometric Analysis (CBA) at JAX at 4 weeks of age. Nuclear magnetic resonance (NMR) was used to assess body composition at 4, 8, 12, 16, and 19 weeks. From weeks 9–13, animals were subject to a series of neurobehavioral testing paradigms, including light-dark test, open field, spontaneous alternation with Y-maze, and voluntary wheel running. From weeks 16–17, animals were subject to 5-day indirect calorimetry trials and intraperitoneal glucose tolerance testing (GTT). At week 18, mice underwent an unconscious EKG to assess cardiac rhythm and function. Mice exited the phenotyping pipeline at 19 weeks. Blood and plasma samples were used to assess multiple biochemical traits and multiple organs were harvested and weighed. A subset of dissected tissues were frozen for future molecular analysis and others were paraffin embedded for future histological study.

trends are again observed using a linear mixed effects model comparison strategy: 26.6% and 8.5% of models including sex and interaction terms, respectively, provide improved fit compared to simpler models that exclude these effects (**S18** and **S19 Tables**). Thus, phenotypic variation is ubiquitous across the Nachman strain panel, and many traits vary between sexes in a strain-dependent manner.

The wild-caught mice used to establish the Nachman lines were subject to unique, location-dependent selective pressures in their native environments. Consistent with this legacy of environmental adaptation, we observe systematic phenotype differences across strains as a function of geographic origin (**S21** and **S22 Tables**). Location is a significant factor accounting for variation in 85.16% of phenotypes (Kruskal-Wallis test, $P<0.05$; n = 930/1092; **S21 Table**). We obtain similar results when fitting linear mixed effect models: 61.2% of models that include geographic sample origin as a random effect provide a better fit compared to models excluding this term (**S23 Table**). For the majority of surveyed traits, strains derived from a common location are more phenotypically similar to each other than strains from distinct locations, echoing trends in sequence-level similarity among strains (**Fig 3**).

To visualize the extent of phenotypic diversity across the Nachman panel, we performed a PC analysis of all surveyed phenotypes. The spatial distribution of strains along the first two PCs (26.4% of variance) affirms model-based findings above, revealing loose clusters based on strain geography and differences between the sexes (**S7 Fig**). While strains are more tightly clustered in the genotypic PCA than the phenotypic PCA, there is significant similarity between the two PC matrices (Similarity of Matrices Index = 0.709; $P < 0.001$; **S7 Fig**). This correspondence with the genotypic PC matrix would seem to suggest that the surveyed phenotypes are subject to polygenic control in the Nachman strains.

Below, we highlight results from each phenotype testing paradigm, present estimates of broad sense heritability, and compare the phenotypic variance among the Nachman samples to that present across classical inbred mouse strain panels. We then present results from correlation analyses that cut across multiple phenotype domains. Comprehensive phenotype data are available in **S24 Table**. Strain-level and Strain x Sex-level phenotype means are provided in **S25 Table**. Broad-sense heritability estimates are provided in **S17 Table**. Results from non-parametric Kruskal-Wallis tests of strain effects on each phenotype are provided in **S16 Table**. Two-way ANOVA results (Phenotype ~ Strain * Sex) are presented in **S20 Table**. Results from

linear mixed effects model fitting are provided in **S18 and S19** **Tables**. Results from Kruskall-Wallis tests, one-way ANOVA, and linear mixed effects models testing geographic origin as an explanatory variable (as opposed to strain) are provided in **S21**, **S22**, **and S23** **Tables**, respectively.

**Body composition analysis by NMR.** We assessed multiple metrics of body composition (body weight, fat mass, lean mass, water mass) via NMR at 5 timepoints between 4 and 19 weeks of age. As expected, all mice gained weight during this period, although the percentage increase in body mass differed among strains across this 15-week interval (range: 28.1% SARC males to 95.92% in GAIC/NachJ males). Males trended toward larger body mass than females (average sex dimorphism across strains at 19 weeks: 4.45 g; **Fig 7**), and all Nachman lines are smaller than C57BL/6J control mice. The broad sense heritability of body weight at 19 weeks is ~0.8 in both males and females (**S17 Table**), indicating that significant proportion of strain variation in body mass is attributable to underlying genetic differences between strains.

We similarly document significant strain and sex differences in body fat mass, lean mass, and water mass (**S16**, **S17**, **and S20** **Tables** and **Fig 7**). Body composition phenotypes are highly correlated over time within strains (average Spearman's rho = 0.669; **S26 Table**), implying that increased body mass is driven by coordinated changes in fat, lean, and water mass rather than isolated changes in one compositional component. Body composition measures

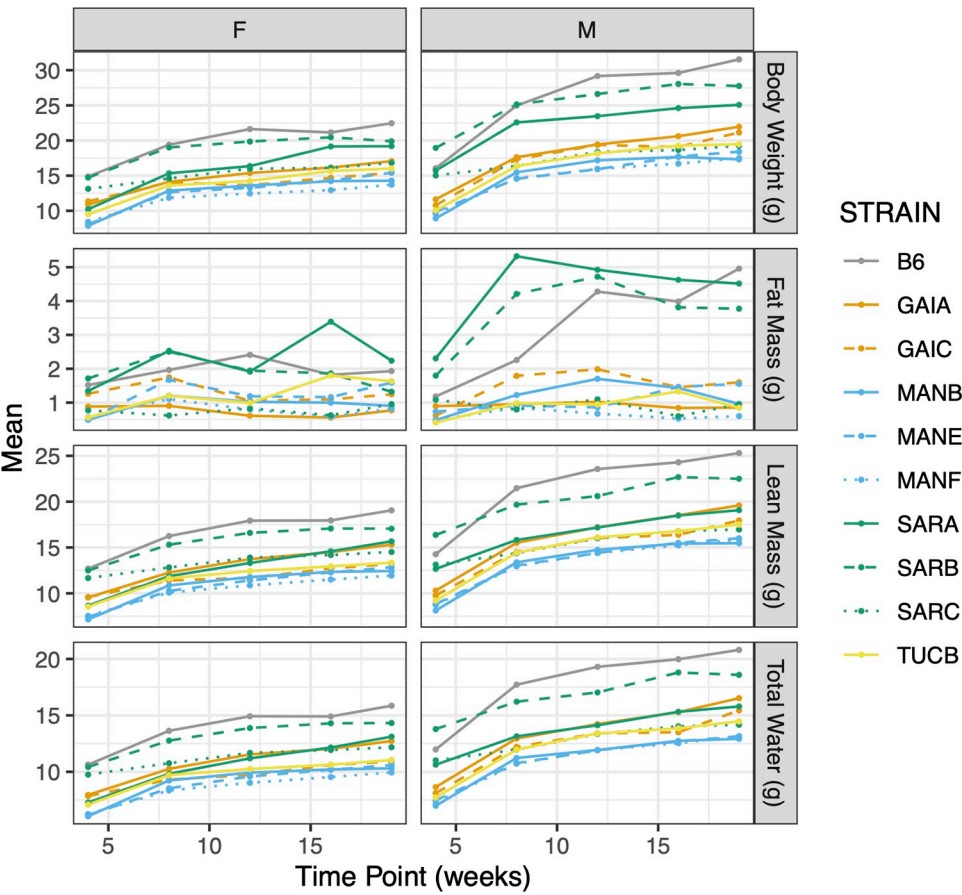

**Fig 7. Changes in body composition over time as assessed by NMR.** Points correspond to strain-level means and data are partitioned by sex to highlight the degree of sex-dimorphism in body composition. Strains are color-coded by geographic origin, with strains from a common location distinguished by line type.

are also correlated across strains at individual timepoints (all Spearman's rho > 0.3; **S7 Fig**), consistent with the observation of near parallel growth trajectories across strains (**Fig 7**).

**Frailty assessment of overall body condition.** Overall health and body condition were assessed using a 29-dimension frailty index score [57]. All strains have low mean frailty index scores (values ranging from 1.75–3.0 out of a possible score range of 0–28; **S25 Table**), although the qualitatively small strain differences observed nearly exceed chance expectations (Kruskal-Wallis $P$ = 0.055; **S16 Table**). Body temperature differed significantly across strains, with GAIC/NachJ and the three strains from Saratoga Springs, NY exhibiting the highest values (Kruskal-Wallis; $P$ = 0.00077; **S16 and S25 Tables**).

**Light-dark test.** The light-dark test is premised on the natural aversion of rodents to being in brightly lit spaces; the proportion of time mice spend in the brightly lit versus dark zones of the testing chamber yields quantifiable metrics of anxiety-like behaviors. Strain TUCB/NachJ shows the lowest overall ambulatory time (24.5 ± 3.54 s; range of other strain means: 30.2–35.8 s), fewest crossovers from one zone to the other (30.9 ± 12.2; range for other strain means: 31.8–61.7), and the greatest proportion of time in the dark chamber zone (67.1%; range for other strain means: 48.0–59.5%; **S25 Table**), consistent with higher overall levels of anxiety-like behavior in this strain. MANE/NachJ spent the greatest proportion of time in the light zone (52%), whereas GAIA/NachJ traveled the greatest total distance in the light zone (819 ± 159 cm; range of other strain means: 436–745 cm). Overall, females are more active in the dark than males, although this trend appears to be largely driven by an especially pronounced sex dimorphism and high levels of activity in SARA/NachJ females (**S9 Fig**). Importantly, the time spent at rest and number of bouts of movement in the dark are moderately heritable ($H^2$ > 0.2; **S17 Table**), revealing a genetic component to observed strain variation.

**Open field assay.** Many measures of locomotion and exploration exhibit high broad sense heritability in the Nachman lines (**S17 Table**). Strains vary more than 3-fold in total distance traveled within the open field arena over the 60-minute test period, ranging from a low of 5158 ± 803 cm in GAIC/NachJ to a high of 17141 ± 4135 cm in SARC/NachJ ($F_8$ = 24.14; $P$ = 6.05x10$^{-24}$). We observe significant sex effects on several open field metrics, including the number of independent movement episodes and vertical activity time (**S20 Table**). Females exhibited more bouts of movement (females: 661 ± 86.6 episodes; males 619 ± 112 movement episodes; $F_1$ = 7.19; $P$ = 0.0082), whereas males spent more time moving in the vertical plane (males: 343 ± 209 s; females: 290 ± 161 s; $F_1$ = 4.12; $P$ = 0.044). We report a significant interaction between sex and strain for the total distance traveled in the center of the arena ($F_8$ = 2.41; $P$ = 0.018; **S20 Table**), with SARA/NachJ, SARB/NachJ, and SARC/NachJ females traversing significantly greater distances than their male counterparts (Wilcoxon Rank Sum Exact Test; $P$ < 0.05). Taken together, these analyses reveal striking strain and sex differences in locomotion and exploration, and reinforce population-level differences in activity levels in the wild [32].

**Spontaneous alternation in the Y-maze.** The spontaneous alternation assay capitalizes on the natural tendency of mice to explore novel environments and serves as a test of spatial working memory. Nachman strains vary in the total number of arm entries and number of spontaneous alternations in the Y-maze, with both phenotypes also exhibiting significant sex and strain × sex effects (two-way ANOVA, $P$ < 0.05; **S20 Table**). The percent alternation for most strains exceeds that for C57BL/6J, suggesting that the Nachman lines have stronger working memories and/or a more intense innate desire to explore novel spaces than C57BL/6J (**S25 Table**). Overall, strains derived from Saratoga Springs, NY exhibit increased percent alternation compared to mice from Manaus, Brazil (58.33 vs. 49.01; Wilcoxon rank sum test, $P$ = 1.25 x 10$^{-6}$; **S21 Table**), a finding consistent with the discovery of increased exploratory behavior in mice from the former location compared to the latter in the open field assay.

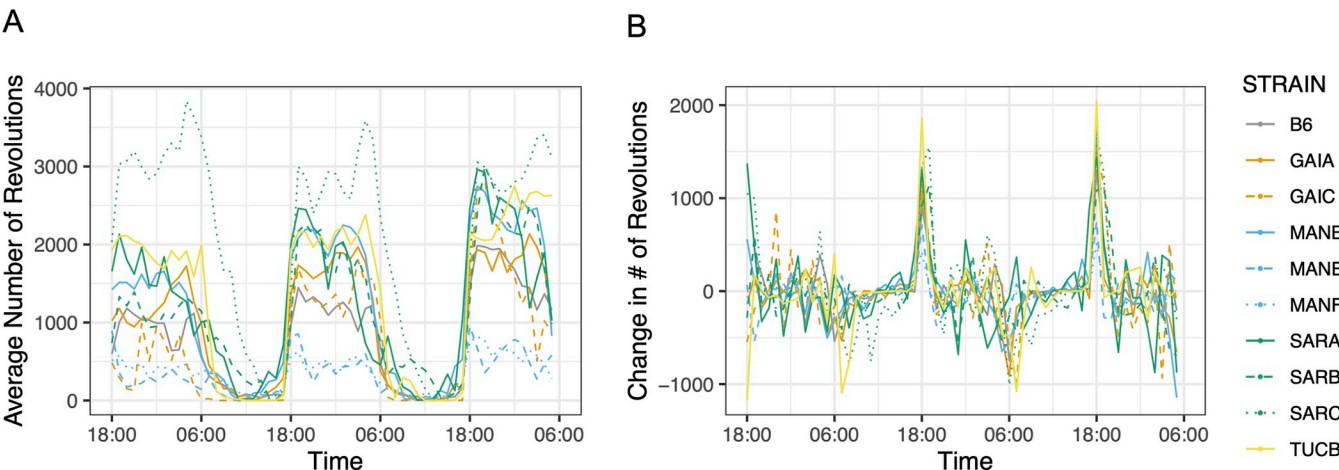

**Fig 8. Strain variation in voluntary exercise and circadian rhythm.** (A) The average number of wheel revolutions per strain over a 60-hour trial period. (B) Hour-to-hour change in wheel running activity (number of wheel revolutions). Abrupt changes in activity correspond to transitions between sleep-wake cycles. In both figures, strains are color-coded by geographic origin, with strains from the same location further distinguished by line type.

**Home cage wheel running.** At week 13, mice were subjected to a 3-day home cage wheel running assay to assess overall activity and circadian rhythms. Strains vary 6.8-fold in total distance run over the 60h trial, from a low of 9,112 m in MANE/NachJ to 62,019 m in SARC/NachJ (**S25 Table**). Mice are nocturnal and, as expected, activity was highest during nighttime hours for all strains (Wilcoxon Rank Sum Test, P < 0.0005; **Fig 8A**).

The Nachman strains derive from wild-caught mice exposed to distinct photoperiods in the wild. We sought to determine whether strains differ in the timing of their peak wheel running as a function of geographic origin. To this end, we computed the change in the number of wheel revolutions per hour across the full trial, defining the transition between day and night as the point exhibiting the most extreme difference in activity level. All strains commence nighttime levels of activity at ~18:00h, but the onset of daytime rest periods is considerably more variable, with strain differences in the duration of nighttime exercise and daytime activity. For example, mice from Manaus, Brazil exhibit a shorter duration and reduced levels of nighttime activity compared to strains from Saratoga Springs, New York, with mice from the latter strain remaining active during much of the day (**Fig 8B**).

**Indirect calorimetry.** Mice were subject to continuous, high-definition respiratory monitoring in Promethion Core cages for 5 days, allowing estimation of energy expenditure, activity levels, and food consumption. We observe significant strain effects on all surveyed phenotypes (P < 0.01), with approximately half of the metabolic traits captured in this assay also exhibiting significant differences between the sexes (**S16**, **S17**, and **S20** Tables). Both sexes of all strains were most active, exhibited highest energy expenditure, and consumed the most food during the dark cycle (**Fig 9A–9C**; Paired Wilcoxon Signed Rank Test, all $P < 10^{-10}$). Most metabolic traits are highly heritable (**S17 Table**), nominating the Nachman lines as excellent models for genetic studies of metabolism. Intriguingly, despite their close genetic affinity, the two strains from Gainesville, Florida exhibited the lowest (GAIC/NachJ; 7.53 kcal/24h) and highest (GAIA/NachJ; 9.98 kcal/24h) total energy expenditure, with C57BL/6J controls presenting an intermediate phenotype (9.67 kcal/24h (**Fig 9D**).

**Glucose tolerance test.** All Nachman lines show an attenuated response to intraperitoneal injection of a controlled concentration of glucose relative to C57BL/6J (**Fig 10**). Peak blood glucose concentrations are higher in males than females for all strains, with exceptionally

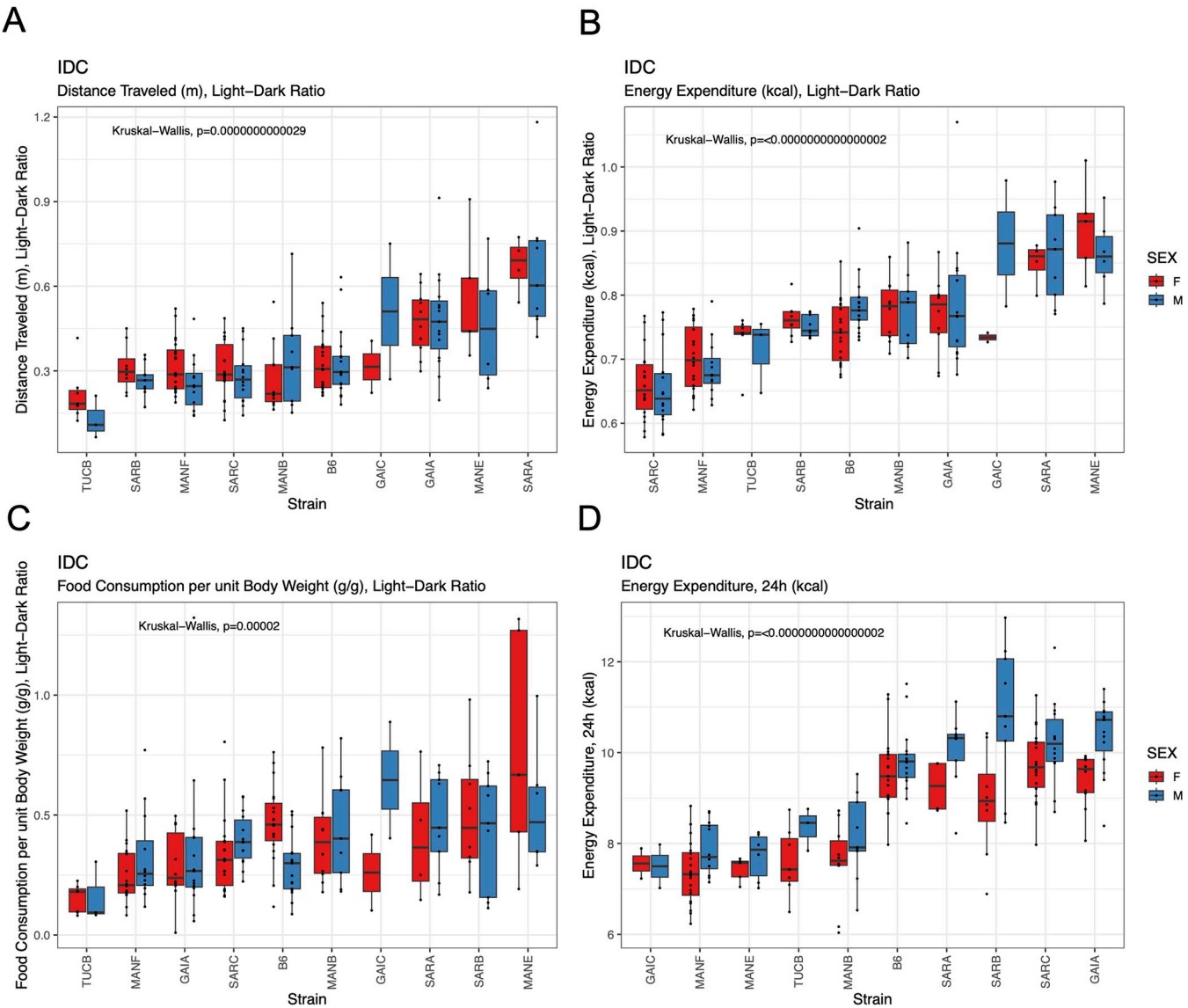

**Fig 9. Strain differences in metabolic phenotypes quantified by continuous respiratory monitoring.** (A) All strains exhibit higher activity during the night, although the extent of the nighttime activity bias varies across strains. (B) Similarly, all strains expend more energy and (C) consume more food at night, but the ratio of day:night energy expenditure and food intake varies across strains. (D) Total energy expenditure varies significantly across strains, with the two strains from Gainesville, Florida (GAIC/NachJ and GAIA/NachJ) delimiting the extremes.

pronounced dimorphisms in SARB/NachJ and GAIC/NachJ (range of sex dimorphism across strains at 15 minutes post injection: 5.29–91 mg/dL). Glucose response curves are significantly different for all pairwise strain comparisons involving SARC/NachJ males and females (Permutation test $P < 0.005$, **S27 Table** and **Fig 10**), emphasizing the unique response profile of this strain. The area under the glucose response curve varies 2.4-fold across strains, with strains GAIC/NachJ and C57BL/6J delimiting the lower and upper extremes, respectively (**S25 Table**). Overall, results from glucose tolerance testing nominate several Nachman strains (most notably, SARC/NachJ) as potential models of Type II diabetes resistance.

**Unconscious electrocardiogram.** Mice underwent unconscious electrocardiograms (EKGs) at week 18 of the phenotyping pipeline (**Fig 6**) to assess heart rate and rhythm. Strains

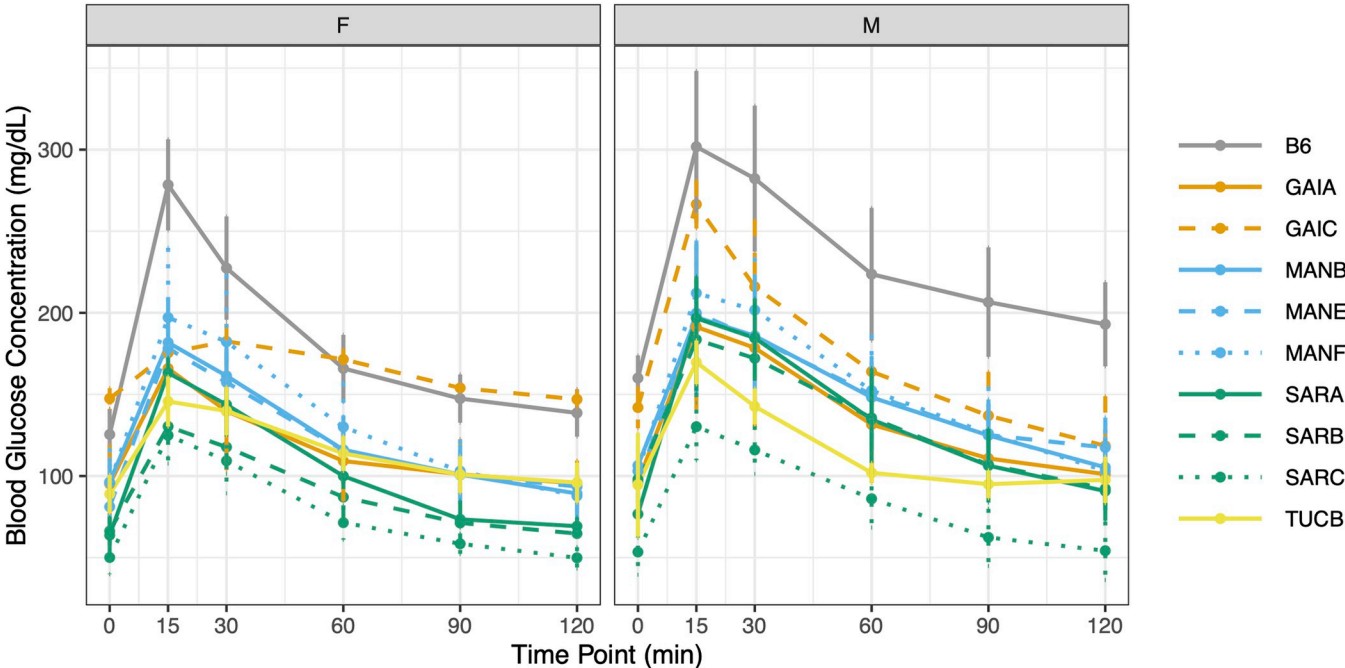

**Fig 10. Glucose response curves over a 2-hour window.** Females and males were analyzed separately. Mice were fasted for 4 hours prior to administration of a fixed (w/v) amount of glucose by intraperitoneal injection. Blood glucose levels were assessed at 15, 30, 60, 90, and 120 minutes post injection. In both panels, strains are color-coded by geographic origin, with strains from a common location further distinguished by line type. Vertical bars denote +/- 1 SD around the strain mean.

exhibit significant variation in all surveyed EKG metrics, including heart rate, peak amplitudes, wave durations, and peak intervals (**S16 Table**; Kruskal-Wallis $P < 0.005$). Nachman lines have lower heart rates, smaller PR intervals, and reduced Q amplitudes than C57BL/6J control animals (**S25 Table**), suggesting clinically relevant differences in cardiac function between laboratory and wild mice.

**Gross morphology and organ weights.** Nachman strains exhibit significant, heritable variation in body length and tail length (Kruskal-Wallis P $< 10^{-8}$; $H^2_{body} = 0.71$; $H^2_{tail} = 0.31$; **S16** and **S17 Tables** and **S10 Fig**). Body length exhibits a significant sex effect, with males having longer bodies than females (**S20 Table**). The magnitude of the sex dimorphism for this morphology trait varies by strain, with SARA/NachJ and SARB/NachJ males exhibiting much longer body lengths than their female conspecifics and only a modest length dimorphism between MANB/NachJ males and females (**S10 Fig**). Similarly, we observe significant heritable variation in organ weights among strains and between the sexes (**S17 Table**). Brain size exhibits exceptionally high heritability ($H^2 = 0.73$) and variability among strains, even after standardizing by total body weight (Kruskal-Wallis $P = 2.94 \times 10^{-9}$), indicating that observed brain size differences among strains are not simply proportional to overall body size.

**Clinical chemistry.** We assessed levels of numerous blood-based clinical markers after subjecting mice to a 4 hr fast at 19 weeks (**Fig 11**). Nachman strains vary in immune cell populations, blood lipid profiles, measures of liver function, and both platelet and red blood cell composition (**S16** and **S17 Tables**). For example, total cholesterol levels range 2.7-fold among strains, with TUCB/NachJ and GAIC/NachJ mice defining the extremes (71.2 mg/dL– 195 mg/dL; **S25 Table**). Males from all strains have higher total cholesterol values than their female conspecifics (average dimorphism = 34.3 mg/dL), a trend echoed in levels of HDL cholesterol (**Fig 11A and 11B**). Nachman lines show exceptionally high variability in platelet counts

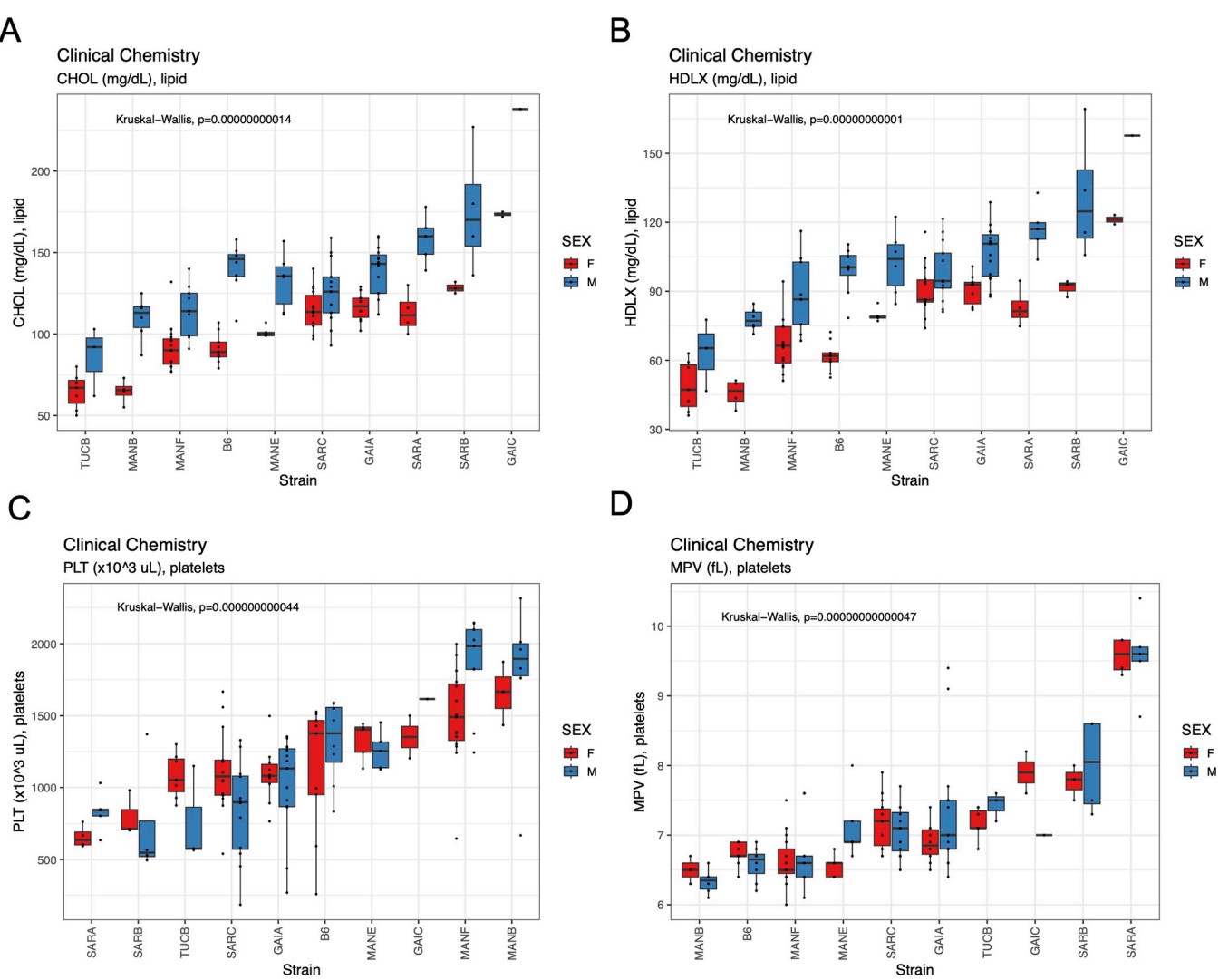

**Fig 11.** Boxplots revealing the extent of strain and sex variation in (A) total cholesterol, (B) HDL cholesterol, (C) platelet count, and (D) platelet volume.

(range of strain means: 754–1725 x10³ cells/uL) and platelet volume (range: 6.39–9.58 fL), with lines from Manaus, Brazil presenting high numbers of low-volume platelets, and strains from Saratoga Springs having low numbers of high-volume platelets (**Fig 11C and 11D** and **S21 and S22** Tables). Multiple immune cell populations, including monocytes, lymphocytes, neutrophils, and eosinophils, exhibit heritable variation across the Nachman strain panel (**S17 Table**), observations that may underlie recent reports of strain differences in pathogen response [58].

## Integrated analysis of Nachman and classical inbred strain phenotypes expands the variance observed in classical inbred strains

We intersected phenotype data from the Nachman lines with existing strain survey datasets deposited on the Mouse Phenome Database (see Methods; [59]). The Nachman animals expand the range of phenotype values observed across inbred strain panels for several phenotypes, including total platelet counts, percent fat mass, and hematocrit levels (**S11 Fig**).

The inclusion of C57BL/6J mice as controls in our phenotyping cohorts provides a common touchpoint between our data and previously published datasets. While we find that C57BL/6J trait values are generally stable across our phenotyped cohorts (**S28 Table**), C57BL/6J trait values differ significantly between our data and legacy datasets (**S29** and **S30** Tables). The inconsistency in trait values could owe to differences in animal age at the time of phenotyping, differences in housing conditions, animal handling by different technicians and research staff, differences in phenotyping protocols or equipment, or other vagaries of the experimental environment. Regardless of the source of this phenotypic variance, it poses major obstacle to phenotype data integration with the Nachman strains and underscores the need for caution in the interpretation of absolute differences in trait values across independently collected datasets.

## Strain-level trait correlations across phenotype assays

Significant strain-level phenotype correlations may reveal traits regulated through shared genetic pathways. As expected, measures of body composition, body size, and organ weights are positively correlated, suggesting a general pattern of isometric growth (**S8 Fig**). Additionally, measures of total activity in indirect calorimetry and open field assays are positively correlated (**S15 Table**). Total serum bilirubin levels are negatively correlated with several phenotypes assessed from the spontaneous alternation assay, including total number of arm entries (rho = -0.97; $P = 0.00017$) and the number of spontaneous alternations (rho = -0. 83, $P = 0.0083$), but suggestively positively correlated with the percent alternation (rho = 0.62; $P = 0.086$). This latter result supports published reports of a negative association between serum bilirubin concentration and cognitive impairment in schizophrenia in humans [60], tentatively extending this correlation to mice. Leptin levels are positively correlated with fat mass at each timepoint assessed by NMR (4, 8, 12, 16, and 19 weeks; rho > 0.66; $P < 0.06$) and gonadal fat pad weight (rho = 0.63; $P = 0.076$), and are negatively correlated with food consumption per unit body weight (rho = -0.85; $P = 0.0061$). These results reinforce the well-established link between leptin secretion and satiety sensing. The duration of the QRS interval on an unconscious EKG is negatively correlated with percent lean mass, but positively correlated with fat mass across strains (**S15 Table**). These findings lend independent support to a recently published association between QRS duration and BMI in humans with no overt cardiac disease [61]. A comprehensive summary of all pairwise trait correlations is provided in **S15 Table**. Sex-specific trait correlations are provided in **S31** and **S32** Tables.

## Discussion

Here, we introduce a new inbred mouse strain resource: the Nachman wild-derived inbred strain panel. Strains in this panel derive from wild-caught mice from unique environments across North and South America. Integrated analysis of the genomes of these new strains with CI strains reveals millions of novel genetic variants segregating in this panel, including predicted deleterious alleles and gene-spanning structural variants. Paralleling this genetic diversity, Nachman strains capture considerable phenotypic variation across biochemical, neurobehavioral, physiological, morphological, and metabolic trait domains.

We integrated our phenotype data with publicly available phenotype datasets from prior laboratory mouse strain surveys to assess phenotypic variability across the Nachman panel within the context of that observed in CI strains. Importantly, the C57BL/6J controls included in our phenotyping cohorts provide a common data point to anchor our data to previously published phenotyping efforts in mouse. However, due to differences in animal ages, experimental methodology, and environmental housing conditions, the vast majority of phenotypes

present inconsistent trait measurements across C57BL/6J controls, limiting the extent to which we can make reliable comparisons across independently surveyed strain panels. These challenges underscore well-known difficulties with data integration and emphasize the importance of standardized phenotyping protocols and detailed metadata reporting [59]. While we acknowledge these caveats, our analyses nonetheless place high certainty on the conclusion that the Nachman strains extend the range of trait values realized using CI strains alone (**S11 Fig**).

The Nachman strain panel complements existing diverse mouse platforms, such as the BXD [62], CC [22], HMDP [63], and DO [23], offering a new community resource for profiling phenotypic variation and assessing responses to exposures, interventions, or treatments. While the 11-member panel is too small to power genetic mapping studies, experimental crosses between Nachman strains with extreme phenotypes can be employed to generate F2, backcross, or advanced intercross lines for experimental mapping. Similarly, crossing knockout alleles into the Nachman strains could enable the identification of naturally segregating modifiers of disease penetrance, severity, or onset.

At the same time, a key feature that distinguishes the Nachman panel from other mouse diversity resources is its exclusive profile of the natural genetic and phenotypic variability in *M. musculus*. During the creation of these WDIS, we aimed to minimize the impacts of artificial selection by randomly selecting breeders at each generation, ensuring that the resultant fixed strain genomes immortalize haplotypes that resemble those found in the wild. In contrast, CI strains are products of strong selection for morphological traits of interest, increased reproductive output, and behavioral phenotypes that promote ease of handling. Such strong artificial selection has almost certainly had broad genomic consequences on unrelated phenotypes through pleiotropy and linkage [64]. This recognition implies that the multigenic architecture of complex trait variation in CI strains may not accurately model that found in natural populations, a consideration that may limit their translational relevance to humans.

Instead, we assert that the Nachman strain genomes mirror multiallelic patterns of diversity that closely approximate the polygenic architecture of traits in nature and, therefore, more accurately model the complex genetic basis of human disease-related phenotypes than CI strains. Beyond the absence of overt artificial selection during strain development, two additional lines of evidence support this assertion. First, our phenotyping efforts highlight the legacy of local adaptive pressures on the organization of phenotype diversity across the Nachman strains. Indeed, we find that geographic origin explains a significant proportion of the variance across strains for the majority of surveyed phenotypes in the Nachman strain panel. For example, we find that SAR mice are typically bigger, more active, and have higher energy expenditure than MAN mice, consistent with predictions of Bergmann's Rule [65] and earlier reports of adaptative phenotypic evolution to colder environments in wild mouse populations [32]. These observations broadly parallel findings from humans, where local adaptive evolution has acted as a major driver of phenotypic divergence between populations [66]. Thus, natural selection has played an important role in sculpting phenotypic diversity in both humans and the Nachman strains. Second, due to their origins from fancy mouse populations purpose-bred for traits of interest, CI strain genomes are mosaics derived from three principal house mouse subspecies [6]. This subspecies composition falls in striking contrast to the structure of wild mouse genomes, which typically bear subspecies ancestry contributions from one, and more rarely two, subspecies [56,67]. Our analyses of subspecies affiliation and introgression reveal that Nachman strains from four of the five sample locations (Manaus, Brazil; Saratoga Springs, New York; Gainesville, Florida; and Edmonton, Alberta, Canada) are of pure *M. m. domesticus* ancestry, but point to moderate levels of introgression from *M. m. castaneus* in mice from Tucson. This adds to prior observations of *M. m. castaneus* introgression into mouse populations from the West coast of North America [68] and earlier SNP-based genetic

investigation of mice from the American southwest [69]. Overall, Nachman strain genomes encapsulate subspecies diversity in a more naturalistic way than current CI strains, maximizing the biological relevance of the phenotypic diversity encoded by their genomes.

Our initial phenotypic and genetic characterization of the Nachman strains provides the basis for many new findings and establishes a powerful suite of resources to spur future research efforts. However, our work presents an initial survey that only scratches the surface of possible biological discovery in this strain panel. For one, all phenotyping was performed "at baseline" using animals fed a standard rodent diet, housed under standard conditions, and in the absence of any experimental treatments. Experimental designs that employ environmental perturbations, different exposure or treatment regimes, aged animals, or other deliberate manipulations may uncover novel phenotypic responses or resilience/susceptibility traits. Second, our phenotyping pipeline was purposefully designed to survey a broad range of biomedically relevant traits and is far from comprehensive. For example, surveyed phenotypes exclude sensory perception phenotypes, bone density, histopathology assessments, and microbiome composition. Deeper, more exhaustive phenotyping of specific trait domains could unlock strain-level differences that fail to manifest using coarser phenotyping assays. Third, due to COVID-related impacts, mice from several strains were poorly represented (GAIC/NachJ, TUCA/NachJ, TUCB/Nach) or altogether absent (EDMB/NachJ) from our phenotyped cohorts (**S14 Table**). As a result, the extent of phenotypic variation across Nachman strains presented here is potentially underestimated–a possibility that future investigations could readily assess. Fourth, we sequenced each strain to modest coverage and relied on comparisons to the GRCm39 reference genome for variant discovery. We obtained only low-coverage of the X and Y chromosomes in our sequenced males (~5x), leading us to exclude these chromosomes from our genomic analyses. Ultimately, we endeavor to perform additional long- and ultra-long read sequencing, generate high-quality *de novo* sequence assemblies, and perform assembly-based variant calling to more comprehensively catalog variation in these strain genomes.

We also acknowledge practical caveats to the use of mice in this strain panel. Our colony breeding records demonstrate that the Nachman strains are robust breeders, but breeding performance falls short of what is typically observed for many CI strains that have been subject to intense artificial selection for productivity (**Fig 2**). Further, like many other WDIS, breeder dams from the Nachman lines do not possess externally visible mating plugs, potentially due to their rapid loss or re-absorption following mating. This may pose challenges for experiments that hinge on the ability to precisely time matings or obtain fetuses at specific gestational ages. Males from many strains also exhibit high levels of aggression toward cage mates and may require single housing or additional enrichment to minimize aggressive encounters. Anecdotally, we have found that supplying mice with wooden blocks for gnawing minimizes tail biting among cage mates. Finally, strains in the Nachman panel retain very high levels of wildness, imposing challenges to routine handling. Of special note, juvenile mice from the TUCB/NachJ strain are remarkable jumpers, easily capable of escaping cages topped with cage extenders (total height ~14"). While working with wild mice may be daunting at first, it is our experience that competency and confidence build quickly. Specialized workstations can aid in containing mice during routine handling, and innovations in animal housing could eventually allow for hands-off cage changes (*e.g.*, via attachment of open plastic tubing to closeable ports on dirty and clean cages, allowing mice to move into new cages on their own). Further, continuous home-cage monitoring coupled with machine-learning based quantification of phenotypes from video footage could obviate many traditional phenotyping paradigms that rely on animal handling and restraint, while also capturing animal behavior in a more ethologically relevant setting [70,71].

The Nachman panel is currently part of the NIH-funded Special Mouse Strain Resource at JAX (P40 OD011102), which provides the framework and setting for its long-term mainte- nance and external distribution. Several additional strains are maintained in Dr. Nachman's private colony and we endeavor to import these additional strains to grow current holdings at JAX. Recognizing that the scientific value of this strain resource is directly tied to the availabil- ity of genomic and phenotypic resources, we have made all raw (*e.g.*, fastq files, raw phenotype measures) and derived data (*e.g.*, vcf files, strain phenotype means) presented in this manu- script available in public repositories and as supplemental material (S1–S33 Tables and S1–S12 Figs). Our team is currently engaged in efforts to expand the collection of tools and resources presented here, including *de novo* genome reference assemblies, embryonic stem cell lines, gene expression datasets, and the initiation of an outbred population founded from a subset of 4 Nachman strains. Together, the inbred Nachman strain panel, its planned derived popula- tions, and accompanying 'omics resources are poised to enable new discoveries in the basic, biomedical, and preclinical research spheres.

## Materials and methods

### Ethics statement

All animal procedures were conducted in compliance with animal care protocols approved by The Jackson Laboratory's Animal Care and Use Committee (Animal Use Summary # 18070) and the University of California Berkeley Animal Care and Use Committee (AUP R361-0514 and AUP-2016-03-8548).

### Inbred strain development

Wild house mice were caught in 2012–2013 from five geographic locations (**Fig 1**) using Sher- man live traps baited with oats. To avoid catching closely related individuals, each mouse was trapped at least 500m from every other mouse. Mice were transported to UC Berkeley and held in quarantine during pathogen testing. Mice from each geographic region were randomly paired to create inbred lines which were propagated through brother-sister mating for at least 10 generations. Initially, 10 independent lines were established from each location. No attempts were made to rescue lines that exhibited infertility due to inbreeding depression, ensuring that surviving lines likely harbor low deleterious mutation loads. The founders of these strains were prepared as museum specimens and have been deposited in the collections of the UC Berkeley Museum of Vertebrate Zoology.

### Mouse breeding, rederivation, and husbandry

Mice from 20 incipient inbred strains were imported from the Nachman Laboratory mouse colony at UC Berkeley to the Jackson Laboratory Importation Facility (**S2 Table**). Strain colo- nies were expansion bred to obtain cohorts of >20 breeding-aged females for oocyte collec- tion. To expedite breeding, strains were mated with some allowance for breeding between generations, a departure from strict sib-sib inbreeding.

Females were superovulated according to standard procedures [72] and harvested oocytes were *in vitro* fertilized with sperm from conspecific males [73]. Embryos were cultured to the 2-cell or 8-cell stage before implantation into pseudopregnant QSi5/IanmTaftJ x C57BL/6J (JAX strains 027001 x 000664) F1 recipient dams of high health status. Live-born pups were transferred to the Dumont Laboratory colony where they were maintained at intermediate health status by strict sib-sib mating. A representative male and female from each successfully

imported strain was genotyped on JAX's standard 54-SNP panel to assure expected strain identity and safeguard against contamination with known laboratory stocks.

Mice were housed under SPF conditions in sterile plastic caging and subject to biweekly cage changes. All animals were fed 6% sterilized rodent chow (Lab Diet ® Formulation 5K0G) and provided with acidified water *ad libitium*. Cages were supplied with aspen bedding material and enriched with nestlets, crinkle paper nesting material, red igloos, and cedar blocks for gnawing.

## Cytogenetic chromosome analysis

Spermatocyte cell spreads were prepared from testis tissue of males aged >8 weeks and immunostained as previously described [74]. Antibodies used were a polyclonal antibody against mouse SYCP3 (1:100 dilution; Novus Biologicals, cat. # NB300-231) and human anti-centromere protein (1:100 dilution; Antibodies Incorporated, cat. # 15–235). Cells were then imaged on a Leica DM6B upright fluorescent microscope equipped with fluorescent filters, LED illumination, and a cooled monochrome Leica DFC7000 GT 2.8 megapixel digital camera. A minimum of 20 cells were captured per strain. Fluorescent intensity and background signal were manually adjusted, and the number of chromosomes counted using ImageJ (v 1.53k).

## Whole genome sequencing

DNA extraction, library preparation, quality control, and sequencing were performed by the Genome Technologies Scientific Service at The Jackson Laboratory. An initial survey of short-read Illumina whole genome sequences from three strains (SARA/NachJ, SARB/NachJ, MANB/NachJ) pointed to high levels of structural variation relative to the mm39 reference genome (S12 Fig). This discovery motivated our use of long-read PacBio HiFi technology for all subsequent sequencing.

High molecular weight DNA was isolated from spleen tissue of a single male from each of the 11 imported Nachman wild-derived inbred strains using the Wizard DNA Purification Kit (Promega) or the Monarch HMW DNA kit (NEB) according to manufacturer's instructions. DNA concentration and quality were assessed using the Nanodrop 2000 spectrophotometer (Thermo Scientific), the Qubit 3.0 dsDNA BR Assay (Thermo Scientific), and the Genomic DNA ScreenTape Analysis Assay (Agilent Technologies). DNA quality from all samples was assessed to be high (260/280 > 1.79 and 260/280 < 1.86, 260/230 > 1.99) and suitable for input for PacBio HiFi library construction.

PacBio HiFi libraries were constructed for each sample using the SMRTbell Express Template Prep Kit 2.0 (Pacific Biosciences) according to the manufacturer's protocols. Briefly, the protocol entails shearing DNA using a g-TUBE device (Covaris), ligating PacBio specific barcoded adapters, and size selection on the Blue Pippin (Sage Science). The quality and concentration of the library were assessed using the Femto Pulse Genomic DNA 165 kb Kit (Agilent Technologies) and Qubit dsDNA HS Assay (ThermoFisher), respectively, according to the manufacturers' instructions. The resultant library for each strain was sequenced on a single SMRT cell on the Sequel II platform (Pacific Biosciences) using a 30-hour movie time.

## Read mapping and variant calling

Read quality and library coverage were assessed using *fastp* [75]. Individual SMRT cells yielded between 22.04 and 38.04 Gb of unique sequence data, with an average read length of 12.9kb (S4 Table). Reads were then mapped to the GRCm39 reference sequence using minimap2 invoking the "HIFI" preset. Per sample single nucleotide calling was performed using Deep-Variant (v1.2.0) under the "PACBIO" model [43]. Per sample gVCF files were then merged using glnexus (v1.2.7) under the DeepVariantWGS configuration to produce a joint call set

[42]. Sites with missing data, genotype quality <30, and indels were subsequently filtered using bcftools (v 0.1.19; [76]). We further eliminated sites with heterozygous calls as these sites are potentially enriched for false positives given our modest sequencing coverage. We refer to this call set as the "Nachman only" call set.

Variants in the Nachman strain panel were compared to those previously discovered in laboratory inbred strains in order to assess the extent to which this new resource captures novel genetic diversity. We downloaded GRCm39-mapped bam files for 51 inbred strains from the European Nucleotide Archive (PRJEB47108). Per sample variant calling was performed using the "WGS" model in DeepVariant (v1.2.0). Joint variant calling of laboratory strain genomes and the 11 inbred Nachman strains was carried out using glnexus (1.2.7) in DeepVariantWGS mode. As above, indels and variants with genotype quality <30 were removed using bcftools (v. 0.1.19). Sites with >10% missing data and heterozygous calls were also excluded. We refer to this callset as the "Sanger-Nachman" call set.

Identical methodology was adopted to generate a joint call set for the Nachman strains and 97 wild-caught *M. m. domesticus*, 22 *M. m. musculus*, 30 *M. m. castaneus*, and 7 *M. spretus*. Sample accession numbers for the wild mouse genome sequences used are provided in **S8 Table**. We refer to this call set as the "wild-Nachman" call set. Again, sites with >10% missing data, genotype quality <30, and indels were excluded using bcftools (v 0.1.19).

Basic statistics were computed on each call set using *bcftools stats*. Call sets were partitioned and intersected using *bcftools view* and and *bcftools isec*, respectively. Variant effects were predicted using the Variant Effect Predictor (Ensembl release 109.3) using the GRCm39 *Mus musculus* assembly [77]. GO Enrichment Analyses were performed using the enrichment analysis tool on the GO Consortium Website (http://geneontology.org/), with the entire set of annotated genes in *Mus musculus* as background [78,79]. Enrichment was determined by a Fisher's Exact test with False Discovery Rate of $P < 0.05$.

## Genome sequence analysis

SNP variation in the Sanger-Nachman call set was summarized via principal component analysis. We first eliminated six strains of non-*M. m. domesticus* ancestry from the call set (CAST/EiJ, CZECHII/EiJ, JF1/MsJ, MOLF/EiJ, PWK/PhJ, and SPRET/EiJ) and removed fixed variants using the *view* command in bcftools (v 0.1.19), invoking the flags *-q 0.01 -s ^CAST_EiJ,CZECHII_EiJ,JF1_MsJ,MOLF_EiJ,PWK_PhJ,SPRET_EiJ*. Variants were greedily thinned to include only those with $r^2 > 0.2$ using PLINK (v1.90b6.18; [80]):

```
plink --vcf $VCF \
 --double-id --allow-extra-chr \
 --set-missing-var-ids @:#
 --vcf-half-call missing \
 --indep-pairwise 50 50 0.2 --out $THINNED_VARIANTS
```

Principle component analysis was performed on the pruned set of variants using the–pca command in plink:

```
plink --vcf $VCF \
 --double-id --allow-extra-chr \
 --set-missing-var-ids @:#
 --vcf-half-call missing \
 --extract $THINNED_VARIANTS
 --make-bed --pca --out $PCA_RESULTS
```

A maximum likelihood phylogenetic tree was constructed from the LD-thinned SNPs on chr19 in the Sanger-Nachman call set using *phyml* (version 3.3; [81]). We focus on chr19 for

the sake of computational efficiency. We first created a fasta format file from the thinned chr19 SNPs using a custom perl script. The output file was then converted to phylip format using the SeqIO.write function in the BioPython (v1.43) library. A maximum likelihood tree was constructed under a GTR model of nucleotide evolution, with nucleotide frequencies computed from the empirical data, and with the transition/transversion ratio, proportion of invariant sites, and gamma distribution of rate classes estimated via maximum likelihood. The executed command was:

```
phyml --input $ALN \
 --datatype nt \
 --bootstrap -1
 --model GTR \
 -f e -t e \
 --pinv e --alpha e \
 --r_seed 12345
```

Identical methodology was used to perform PC analysis and phylogeny construction on the Nachman-wild call set. Prior to analysis, we first filtered the vcf file to include only wild *M. m. domesticus* samples, recognizing that PCA results would be dominated by subspecies-level diversity if samples from *M. m. musculus* and *M. m. castaneus* were retained. We also further subset the Nachman-wild call set to exclude *M. m. domesticus* samples from Iran.

## Introgression analysis

Nachman strain genomes were scanned for potential signatures of introgression from *M. m. castaneus* and *M. m. musculus* using the allele sharing methods implemented in *Dsuite* (v. 0.5) [82]. Wild *M. m. domesticus* and *M. m. castaneus* (or *M. m. musculus*) genomes in the wild-Nachman call set were utilized as P1 and P3, respectively, with *M. spretus* genomes specified as the outgroup taxon (**S8 Table**). *Dsuite Dtrio* was run separately on each chromosome and with each inbred Nachman strain individually profiled as P2. Chromosome-level statistics were integrated into genome-wide estimates using *Dcombine*. To pinpoint specific sites of introgression between *M. m. castanus* and each focal strain, a windowed analysis was performed using the *Dinvestigate* command in *Dsuite* with a window size of 5000 informative sites and 2500 site slide. Output statistics were plotted in RStudio (2022.02.0, Build 443). We focus on sites with an excess of derived alleles shared between each of our profiled Nachman lines and *M. m. castaneus* and extracted the 5% of windows with the highest $f_{dM}$ estimates consistent with potential introgression from *M. m. castaneus*. These outlier windows were then intersected with Ensembl genes (v. 109; *Mus musculus* GRCm39) using bedtools *isec* (v. 2.28.0; [83]) and subjected to a GO Enrichment Analysis using the online enrichment analysis tool on the GO Consortium Website (http://geneontology.org/) [78,79]. We specified the entire set of annotated genes in *Mus musculus* as background to identify specific biological processes enriched in putative regions of *M. m. castaneus* introgression. Enrichment was determined by a Fisher's Exact test with False Discovery Rate of $P < 0.05$.

## Structural variant discovery and calling

We identified SVs in the 11 Nachman wild-derived inbred strain genomes using both *pbsv* (https://github.com/PacificBiosciences/pbsv) and *sniffles2* (v. 2.0.7; [84]). *pbsv* was first run on each sample in *discover* mode to identify read signatures consistent with possible SVs. SVs where then called and samples jointly genotyped by executing *pbsv* in *call* mode. Tandem repeats in the GRCm39 assembly were identified using the findTandemRepeats.py script (https://github.com/PacificBiosciences/pbsv/commit/

bcec7d382f3ea40158ed9cca3c5fef9686a76641) and supplied when executing *sniffles2* to improve the accuracy of calls in repetitive regions. Per sample SV calls generated by *sniffles2* were merged and filtered to include only autosomal calls using *bcftools merge* (v. 0.1.19). Calls with close or overlapping breakpoints across samples were collapsed using *truvari* (v4.0.0; [85]), with the following parameters specified: -pctsize 0.75 –pctovl 0.5 –pctseq 0.7 -s 20 -S 10000000 -k common—chain. We then intersected *pbsv* and *sniffles2* SV calls using *truvari bench* to produce a higher confidence call set. We used the pbsv callset as the "truth" set and invoked the following command line parameters: -pctsize 0.75 –pctovl 0.5 –pctseq 0.7 –dup-to-ins–passonly -sizemin 20. SVs were annotated using the Ensembl Variant Effect Predictor (release 109.3) and gene model annotations from the GRCm39 assembly.

Deletions and insertion sequences in this intersection call set were converted to fasta format and input to RepeatMasker (v 4.1.5). The following repeatMasker command was executed:

```
RepeatMasker \
-e hmmer
-q \
-species "mus musculus" \
-lcambig
-nocut \
-div 50 \
-no_id \
-dir $OUTDIR \
-trf_prgm $PATH_TO_TRF/trf409.linux64
-hmmer_dir $PATH_TO_HMMER/hmmer-3.3.1/src \
$FASTA
```

The number and base coverage of each family and class of transposable element within SVs was calculated from the alignment file output of RepeatMasker using the parseRM.pl script available at: https://github.com/4ureliek/Parsing-RepeatMasker-Outputs/blob/master.

Lastly, SV calls were intersected with a previously published SV call set for a diverse set of inbred mouse strains [49]. Shared SVs (>75% reciprocal overlap) were identified using *truvari bench*, with the Nachman callset featured as the "truth" set and invoking the same command line parameters as above.

## Animal phenotyping

We developed a phenotyping pipeline loosely modeled on that used by the KOMP2 Project (https://www.mousephenotype.org/impress/index), with modifications to account for the overall health and wildness of the Nachman strains (**Fig 6**). Phenotypes broadly profile disease-relevant neurobehavioral, physiological, metabolic, biochemical, and morphological trait variation.

Mice were organized into 16 phenotyping cohorts ranging in size from 6–16 animals (mean = 13.4; **S14 Table**). With the exception of cohort 13, each cohort was composed of animals from multiple strains (N = 2–5 strains) and mice of both sexes. Cohort 13 included only animals from strain GAIA/NachJ. Ten of the 16 cohorts included C57BL/6J mice as controls (n = 2–5 mice per cohort), providing a means for ensuring stability of phenotype measurements across cohorts. A total of 215 mice (108 female, 107 male) were phenotyped from 9 of the 11 Nachman lines in two waves. The first wave included 5 cohorts that were phenotyped between August 2020 and November 2020. The second wave included the remaining 11 cohorts which were phenotyped between June 2021 and October 2021. Strain EDMB/NachJ was not imported in time for inclusion in this phenotyping effort.

All live animal phenotyping was performed by trained staff in the Center for Biometric Analysis at The Jackson Laboratory (JAX). Animals were ~4 weeks (± 6 days) at study intake and exited the phenotyping pipeline at 19 weeks (± 6 days). Terminal collections and tissue harvests were performed by the Necropsy Scientific Service at JAX, with samples subsequently transferred to the Clinical Chemistry and Histopathological Sciences Scientific Services at JAX.

**Body composition assessment by nuclear magnetic resonance.** Body composition analysis was carried out on mice at weeks 4, 8, 12, 16, and 19. Conscious, unrestrained mice were individually placed in an acrylic tube that was inserted into a EchoMRI-3-in-1 Body Composition Analyzer (EchoMRI LLC) to yield non-invasive estimates of mass and percentage estimates of fat, lean, and water mass.

**Frailty assessment (9 weeks).** Mice were gently restrained for visual assessment of 29 metrics of physical condition using a modified version of the protocol described in [57]. Observational assessment of most parameters was qualitatively categorized as 0 if normal, 1 if highly abnormal, and 0.5 if intermediate. Exceptions include body temperature and body weight, which were measured in standard, qualitative units (Celsius and grams, respectively). A general frailty index score was computed by summing the scores for individual parameters (excluding body weight and body temperature).

**Light-dark test (10 weeks).** The light-dark test is a classic behavioral paradigm for quantifying anxiety-related behaviors in rodents [86]. Following ~60 minute acclimation to the testing room, mice were individually placed in a square, plexiglass arena (40 x 40 x 40 cm) divided into two compartments separated by a doorway. One side of the chamber was illuminated by an external light to ~400–450 Lux, while the other side remained darkened. Mouse movements were tracked over a span of 10 minutes via an infrared photobeam 3-dimensional grid system invisible to the animals. Beam breaks were computationally decoded to provide information about latency to enter the lighted side of the chamber, total number of transitions between the light and dark sides, total distance traveled, and time spent in the light versus dark sides.

**Open field test (11 weeks).** Mice were first acclimated to the testing room for ~60 minutes. Animals were individually placed into the center of a square plexiglass arena (40 x 40 x 40 cm) overlaid by a sensitive infrared photobeam three-dimensional grid. Beam breaks were computationally decoded into quantitative measurements of distance traveled, rearing, time spent within certain zones of the arena, and repetitive behaviors over the span of 60 minutes.

**Spontaneous alternation with Y-maze (12 weeks).** Following a 60-minute room acclimation, a single mouse was placed into the center of an opaque Y-shaped testing arena composed of three equally sized arms each measuring 25–35 cm long, 5–6 cm wide, and with walls extending 12-18cm high. Mouse movements were recorded by a camera interfaced with tracking software (Noldus Ethnovision) to quantify distance traveled, arm entries, time spent in each arm, and the number of correct alternations (*i.e.*, a visit to each of the three arms before returning to any arms).

**Home cage wheel running (13 weeks).** Singly housed mice were provided with low profile running wheels (Med Associates) equipped with wireless transmitters for ~3 days. Transmitters recorded the number of wheel revolutions and time on one-minute intervals. Data were forced onto a common timeline delimited by 18:00:00 on Day 1 to 6:00:00 on Day 3, converted to distances based on running wheel diameter (15.5cm), and aggregated over 5 minute, 30 minute, 1h, 6h, and 12 h intervals for analysis.

**Indirect calorimetry (16 weeks).** Mice were acclimated to single-housing for 5–7 days prior to transfer to Promethion Core cages for continuous high-definition respiratory monitoring over a 5-day span. Rates of oxygen consumption and carbon dioxide production were assessed using a built-in respirometry system, allowing calculation of overall energy

expenditure using manufacturer software. Food and water intake were continuously monitored via high precision weight sensors mounted to the cage lid. Cages were overlaid with a grid of high sensitivity infrared beams to track animal activity and locomotion in continuous time.

**Glucose tolerance testing.** Animals were fasted for 4 hours, weighed, and restrained to make an angled incision at the tail for repeat blood collection. An initial blood drop was placed on a glucose test strip and a baseline blood glucose recorded using a handheld glucometer. Animals were then administered sterile glucose solution at a dose of 2g/kg of body weight via intra-peritoneal injection. Blood glucose levels were monitored at 15, 30, 60, 90, and 120 minutes post injection using single drops of blood obtained from the tail incision.

**Unconscious electrocardiogram and gross morphology.** Mice were anesthetized by exposure to isoflurane gas in an enclosed chamber. Once unconscious, each animal was removed from the chamber and placed ventral side up on an ECG platform, with continued gas delivery administered through a nose cone. Paralube ophthalmic ointment was placed on the eyes to prevent drying during testing and recovery. Single-use pre-gelled self-adhesive silver/silver chloride ring recording electrodes (MVAP Medical Supplies, EMG electrodes) were adhered to the palmer surface of both front feet and the plantar surface of the left hind food. Leads were connected to each electrode and attached to a FE132 BIoAmp (AD Instruments) electrical signal amplifier which was fed into a PowerLab 4/35 digital-to-analog converter system and 4-channel recorder (AD Instruments) connected to a laptop computer with LabChart software. Core temperature was monitored and regulated via rectal probe physiology monitoring system and automatic heat pad. The ECG tracing was recorded until 30 seconds of waveform signal was obtained.

While animals remained unconscious, overall length, body length, and tail length were measured using handheld calipers.

**Terminal collections.** Approximately ~400ul whole blood was collected from the submental vein of 4-hour fasted mice. Animals were then euthanized via $CO_2$ asphyxiation in accordance with recommendations from the American Veterinary Medical Association. Blood samples were transferred to the Clinical Chemistry Scientific Service at The Jackson Laboratory for measurement of the following blood-based clinical traits: albumin, alkaline phosphatase, alanine transaminase, aspartate transaminase, blood urea nitrogen, enzymatic creatinine, glucose, total cholesterol, HDLD cholesterol, triglycerides, insulin, leptin, iron, total bilirubin, total protein, and complete blood count with differential. The following organs were carefully dissected from each animal by skilled staff in the Jackson Laboratory Necropsy Scientific Service: skeletal muscle, gonadal fat pad, brown adipose from between the shoulder blades, tail, skin (with hair), femur, spleen, kidneys, liver, brain, gonads, heart, lungs, eye. Tissues were weighed, frozen, and fixed in 10% NBF and paraffin embedded according to the protocol presented in **S33 Table**.

## Phenotype data integration, statistical analysis, and broad sense heritability estimation

All phenotype data were wrangled into a single file in RStudio (v. 2022.02.0 Build 443) using the dplyr (v. 1.1.1), tidyverse (v. 2.0.0), data.table (v. 1.14.8), readxl (v. 1.4.2), and lubridate (v. 1.9.2) packages (**S24 Table**). Categorical traits collected as part of the frailty analysis (n = 27) showed little variation among strains (**S25 Table**) and were excluded from statistical analyses. All remaining phenotypes were continuous. We tested for overall strain effects on each phenotype using non-parametric Kruskall Wallis (**S16 Table**) and one-way ANOVA tests (phenotype value ~ strain; **S17 Table**). Additionally, we tested for an effect of geographic origin on

variation in each phenotype using Kruskall-Wallis (**S21 Table**) and one-way ANOVA tests (phenotype value ~ sample locale; **S22 Table**). C57BL/6J controls were excluded from these analyses to allow specific focus on variation across the Nachman strains. Although the ANOVA assumption of normality does not hold for many analyzed traits, there is strong agreement of *P*-values estimated by these two statistical methods (Spearman's Rho = 0.929, $P < 2.2 \times 10^{-16}$). Given that our objective is to flag phenotypes that likely vary across the Nachman panel (rather than make robust claims about specific phenotypes), reported *P*-values are not corrected for multiple testing.

PC analysis was performed on the set of traits from each phenotype assay using the pca command within the pcaMethods R package [87]. We distilled trait values into per strain, per sex means and normalized the resulting values to have unit variance centered on zero. We then identified the number of PCs required to explain 90% of the variance associated with each phenotype assay and combined these into a single matrix. PC analysis was performed on this combined matrix to summarize multidimensional trait variation across strains and sexes (**S7 Fig**). To compare the phenotypic PC matrix with the genotypic PC matrix, phenotypic PC values were averaged across both sexes within a strain. Matrix similarity was assessed by the similarity of matrices index using the *SMI* function call within the MatrixCorrelation R package [88]. Significance of the resulting SMI value was determined by 1000 permutations.

Broad sense heritability was estimated from one-way ANOVA output by the intraclass correlation method [89]:

$$H^2 = \frac{MSB - MSW}{MSB + (n-1)MSW},$$

where MSB and MSW are the mean squares between and within strains, respectively. Heritability estimates were computed from one-way ANOVA tests run on each sex separately, as well as both sexes combined (**S17 Table**). Two-way ANOVA was also performed to test the impact of strain, sex, and their interaction on the variability observed in each trait (phenotype value ~ strain * sex; **S20 Table**). Spearman's rank correlate was used to quantify trait correlations at the strain, strain × sex, and individual levels.

In parallel to this approach, we also fit a series of nested linear mixed effect models to assess the impact of strain, sex, and strain × sex interaction on each phenotype. Specifically, for each trait *t*, we fit the following series of models using the *lmer* command within the *lme4* (version 1.1–32) R package:

$$Phenotype\ Value_t = Sex + (1|Strain) + (1|Strain : Sex) + (1|Cohort) \tag{1}$$

$$Phenotype\ Value_t = Sex + (1|Cohort) \tag{2}$$

$$Phenotype\ Value_t = (1|Strain) + (1|Cohort) \tag{3}$$

$$Phenotype\ Value_t = Sex + (1|Strain) + (1|Cohort) \tag{4}$$

Sex was treated as a fixed effect, with strain, cohort, and the interaction between strain and sex treated as random effects. Nested models were compared by a log likelihood test using the *anova* function call in R. Specifically, models 1 and 2 were compared to assess whether inclusion of strain as a random effect significantly improved model fit. Models 1 and 3 were compared to assess whether inclusion of sex significantly improved model fit, and models 1 and 4 were compared to determine whether the interaction between strain and sex meaningfully improved model fit. Owing to the modest amount of data and the complexity of these models,

many models were singular, necessitating caution in the interpretation of results from this model fitting approach. Of the 1092 traits analyzed under this mixed model framework, only 358 provided non-singular model fits (S18 Table).

Many phenotypes are highly correlated with body weight (S15 Table), motivating us to also consider body weight-adjusted trait values in our models. We fit the residuals from a simple linear regression of each trait value on body weight at 18 weeks in each of the models above. This had a similar impact on the number of singular model fits (1083 analyzed traits, with 462 with non-singular fits across models 1–4; S19 Table). We excluded body weight traits at all assessed timepoints, accounting for the reduced number of phenotypes analyzed under this framework.

To explore the role of geographic sample location on observed variation across strains, we fit linear mixed effect models analogous to those in Eqs (1) and (2), exchanging the random factor "Strain" for "Location".

$$Phenotype\ Value_t = Sex + (1|Location) + (1|Location:Sex) + (1|Cohort) \tag{5}$$

$$Phenotype\ Value_t = Sex + (1|Cohort) \tag{6}$$

Model comparisons were performed via likelihood ratio test to determine whether inclusion of location as a random effect significantly improved model fit. Phenotype values were first regressed against body weight at 18 weeks to eliminate the correlation with this trait. As above, the majority of phenotypes provided singular fits (S23 Table).

## Comparisons with legacy phenotype data

We accessed the CGDpheno1 and CGDpheno3 datasets from the Mouse Phenome Database [4]. These strain survey datasets were prioritized because: (1) they share multiple phenotypes in common with those measured in the Nachman strains, (2) they were generated within the CBA at JAX, providing methodological consistency in data collection and similarity in housing conditions, and (3) animals were not subject to any experimental treatments. Public datasets were combined with the Nachman data using base R commands and dplyr (v. 1.1.1), with attention paid to consistency of measurement units. Where needed, values were converted onto common unit scales using the appropriate conversion factors or mathematical manipulations.

## Supporting information

**S1 Fig. Male reproductive traits in the inbred Nachman strains, a subset of their F1 hybrids, and representative inbred strains.** All mice were >8 weeks old at time of phenotyping. (A) Testis weight standardized by body weight for inbred Nachman strains and three derivative F1 hybrids. Values for the classical inbred strain C57BL/6J (B6), a sample of DO animals, and a representative subset of CC strains are included for comparison. (B) Sperm density estimates for 8 genetically diverse inbred mouse strains, EDMB/NachJ (EDMB), and 3 F1 hybrids derived from crosses between Nachman strains. Inbred strain sperm density estimates are from the Handel1 dataset on the Mouse Phenome Database.
(TIF)

**S2 Fig. Cytogenetic analysis of spermatocyte cell spreads confirms 2N = 40 karyotypes in Nachman strains.** (A) Table summary of the number of individuals and number of cells analyzed for each of 5 Nachman strains. All cells derive from males and all harbor the standard house mouse karyotype defined by 19 acrocentric autosome pairs and a pair of sex

chromosomes. Representative pachytene-stage spermatocyte cell spreads from TUCB/NachJ (B) EDMB/NachJ (C), MANF/NachJ (D), and GAIC/NachJ (E) stained with antibodies against SYCP3 (red), a component of the meiotic synaptonemal complex that localizes along the paired chromosome axes, and anti-centromere antibodies (blue).
(TIF)

**S3 Fig. Heterozygosity declines as a function of inbreeding generation.** (A) The observed inbreeding coefficient, $F$, was calculated using the method of moments estimator implemented in vcftools (v 0.1.16) [11,90–92]. For most strains, the empirical $F$ estimate closely tracks with the theoretical expectation for increasing inbreeding generation number (black line). This expectation was computed from the recurrence equation: $F_t = 0.25(1 + 2F_{t-1} + F_{t-2})$, where $F_t$ corresponds to the inbreeding coefficient at generation $t$. Strain MANF/NachJ presents an exception that is likely attributable to departures from a strict sib-sib mating design during the colony expansion effort that preceded strain rederivation. (B) Per sample nucleotide diversity ($\pi$) declines with increasing inbreeding generation, as expected.
(TIF)

**S4 Fig. Higher principal component dimensions provide additional refinement to the genetic relationships between the Nachman and CI mouse strains.** (A) Percentage of total variance explained by each PC 1–20. (B) Dot plots of PC3 versus PC4, PC5 versus PC6 (C), and PC7 versus PC8 (D). The color legend in (B) also applies to (C) and (D).
(TIF)

**S5 Fig.** Distribution of deletion (A) and insertion (B) lengths across the Nachman strains. For ease of visualization, only SVs <10kb in length are plotted. Only 49 deletions and 1 insertion call exceed this cutoff. Peaks at ~200bp and ~6kb suggest contributions from SINE elements and full-length L1 elements.
(TIF)

**S6 Fig. Genomic distribution of $f_{dM}$ in TUCA and TUCB.** $f_{dM}$ was calculated in 5000 SNP windows (2500 SNPs slide) using genome sequences from wild-caught *M. m. domesticus*, wild-caught *M. m. castaneus*, and *M. spretus* (**S8 Table**). Dashed lines correspond to the 95th percentile of the most extremely positive $f_{dM}$ statistics for each strain, delimiting regions of likely *M. m. castaneus* introgression.
(TIF)

**S7 Fig. PC analysis on all phenotypes.** Individual PC analyses were first performed on normalized per sex, per strain mean trait values from each phenotyping assay. The minimum number of PCs required to explain >90% of the variance for each phenotype assay were aggregated, and a second PCA was performed on the resulting matrix. Results from this second PCA are plotted in (A), with PC1 (14.5%) and PC2 (11.9%) together accounting for 26.4% of the variance. PC values were averaged over sexes within a strain and axes rotated 180˚ to produce (B). Panel (C) duplicates the results from the genotypic PC analysis presented in **Fig 3B** to enable appreciation of the similarity of the strain distribution pattern along the first two PCs.
(TIF)

**S8 Fig. Correlation heat map for body composition and body weight metrics assayed at 4, 8, 12, 16, and 19 weeks of age.** Numbers in the axis tick labels correspond to animal age at the time of measurement. The magnitude of the correlation coefficient (Spearman's rho) is indicated by the scale bar, with higher correlations denoted by darker red colors, and weaker correlations indicated in yellow.
(TIF)

**S9 Fig. Strain variation in several anxiety-related metrics assessed by the light-dark test.** The centroid position on the test subject's body was used to assign position in either the dark or light halves of the testing apparatus. In (C), distance is plotted on the y-axis in meters. The color legend presented in (C) applies to all panels.
(TIF)

**S10 Fig.** Strain and sex variation in body length (A) and (B) brain weight. Brain weight is standardized by total body weight.
(TIF)

**S11 Fig. Nachman strains expand the range of phenotypic trait variance observed in current inbred strain and mouse diversity panels.** (A) Platelet counts (cells x $10^3$/ul) for the Nachman strains were integrated with the CGDpheno3 dataset from the Mouse Phenome Database. (B) Percent fat mass at 12 weeks in the Nachman lines and strains profiled in the CGDpheno1 dataset. (C) Percent hematocrit in the Nachman lines and strains in CGDpheno3. The color legend in (A) applies to all panels. In each panel, Nachman strains are indicated by red bars along the x-axis and boxplots for C57BL/6J mice are marked by black boxes. C57BL/6J mice phenotyped as controls alongside the Nachman strains are denoted as "C57BL/6J-NACH"; C57BL/6J animals phenotyped in other studies are indicated by the x-axis label "C57BL/6J".
(TIF)

**S12 Fig.** Cumulative distribution of copy number state in 1kb sliding windows across the autosomal genome for (A) SARA/NachJ, (B) SARB/NachJ, and (C) MANB/NachJ. To estimate copy number, read depth was first computed in 1kb sliding windows (no overlap) using *mosdepth* (executed command: mosdepth -n -b 1000 -t 2 -x $PREFIX $BAM). Raw read depth values were then corrected for potential GC-biases introduced during library preparation. Briefly, we used the GRCm39 reference genome to compute the observed GC content of each 1000bp window. GC content values were rounded to the nearest 0.001 and regions with identical GC content were binned. For each strain, we then computed the mean read depth across all genomic windows that fell into each GC content bin. Next, we fitted a second degree polynomial to the relationship between read depth and GC content using the scatter.smooth function in R and with span parameter of 0.7. For each GC-bin, we then computed the difference between the fitted polynomial and the genome-wide average read depth. These values correspond to the magnitude of "inflation" or "deflation" in read depth across windows of a given GC-content due to systematic GC biases in the data. The read depth value in each 1000bp window was then adjusted by the appropriate GC correction factor. Finally, these GC-corrected read depths were divided by the average per-sample coverage to convert into absolute copy number estimates. The cumulative distribution of autosomal copy number estimates was then computed using the *ecdf* function call within the *stats* package in Rstudio (version 2022.02.0, Build 443). The proportion of windows with copy number <1.5 and >2.5 (red vertical lines) was calculated as a proxy for the extent of structural divergence between the focal strain genome and GRCm39.
(TIF)

**S1 Table. Summary of currently available wild-derived inbred mouse strain resources.**
(XLSX)

**S2 Table. Status of Nachman strains imported to The Jackson Laboratory.**
(XLSX)

**S3 Table. Male reproductive trait measures from the inbred Nachman strains, Diversity Outbred mice, Collaborative Cross strains, and common classical inbred strains.** Diversity Outbred data were collected by the Dumont Lab in 2019. Collaborative Cross and inbred strain data were accessed from the Mouse Phenome Database under the following data set names: Handel1, Lazear1, Odet1, Shorter4, and Shorter5. The table is arranged in long-data format. (CSV)

**S4 Table. Whole genome sequencing quality metrics, coverage, and sample information.** (XLSX)

**S5 Table. This workbook contains three spreadsheets that summarize the predicted effects of single-nucleotide variants ascertained in the Nachman strains.** The first sheet ("SNPs") summarizes the number of variants of different predicted functional classes in the Nachman strains. Numbers are presented for all variants in the Nachman strains and the subset of variants that are private to the Nachman strains and not observed in classical inbred strains included in the Mouse Genomes Project. The second spreadsheet ("HIGH Nachman Private") lists SNPs private to the Nachman strains (*i.e.*, not present in Mouse Genomes Project data) with HIGH predicted functional impact. Impacted genes, their associated Gene Ontology terms, and strains carrying the predicted deleterious allele are also included. The third spreadsheet ("GO Analysis") presents the output of a GO enrichment analysis on genes in the "HIGH Nachman Private" spreadsheet against all genes in the *M. musculus* genome. (XLSX)

**S6 Table. Ensemble gene identifiers and gene symbols for transcripts predicted to be ablated by structural variants present in the Nachman strains.** (TXT)

**S7 Table. Repetitive element counts in duplication and deletion structural variant calls in the Nachman strains.** (XLSX)

**S8 Table. ENA and SRA accession numbers for the publicly available whole genome sequences of inbred laboratory strains and wild mice analyzed alongside the Nachman strains.** (XLSX)

**S9 Table. Estimated D-statistics, f4-ratio, and admixture proportions for each tested population trio.** Wild *M. m. domesticus* genomes were utilized as population 1 (P1), with either wild *M. m. castaneus* or wild *M. m. musculus* genomes specified as P3. Each inbred Nachman strain was individually profiled as P2. (XLSX)

**S10 Table. Estimated per window D-statistics for the *M. m. domesticus*–TUCA/NachJ–*M. m. castaneus* trio.** Analyses were performed on windows of 5000 SNPs with a 2500 SNP slide. (TXT)

**S11 Table. Estimated per window D-statistics for the *M. m. domesticus*–TUCB/NachJ–*M. m. castaneus* trio.** Analyses were performed on windows of 5000 SNPs with a 2500 SNP slide. (TXT)

**S12 Table. GO enrichment analysis on genes within the 5% of windows with the highest $f_{dM}$ statistics presented in [S10 Table].** These windows correspond to likely regions of *M. m. castaneus* introgression into TUCA/NachJ. The set of all genes in the *M. musculus* genome was specified as background, with enrichment determined by a Fisher's Exact Test with False

Discovery Rate of $P < 0.05$.
(XLS)

**S13 Table. GO enrichment analysis on genes within the 5% of windows with the highest $f_{dM}$ statistics presented in S11 Table.** These windows correspond to likely regions of *M. m. castaneus* introgression into TUCB/NachJ. The set of all genes in the *M. musculus* genome was specified as background, with enrichment determined by a Fisher's Exact Test with False Discovery Rate of $P < 0.05$.
(XLS)

**S14 Table. Strain and sex composition of the 16 phenotyping cohorts.**
(XLSX)

**S15 Table. Pairwise strain-level Spearman correlations between surveyed phenotypes.** Values were averaged over all phenotyped animals within a strain.
(TXT)

**S16 Table. Results from non-parametric Kruskal-Wallis tests assessing the effect of strain on each surveyed phenotype.**
(TXT)

**S17 Table. Results from one-way ANOVA to quantify the effect of strain on the variability in each surveyed quantitative phenotype.** The broad-sense heritability of each trait was estimated from the output variance components.
(TXT)

**S18 Table. Results from likelihood ratio tests evaluating the fit of nested linear mixed-effects models to assess the significance of strain, sex, and strain × sex interaction terms on the variance associated with each trait.**
(TXT)

**S19 Table. Results from likelihood ratio tests evaluating the fit of nested linear mixed-effects models to assess the significance of strain, sex, and strain × sex interaction terms on the variance associated with each trait.** Trait values were first regressed on body weight at 18 weeks and residuals used for model fitting.
(TXT)

**S20 Table. Results from two-way ANOVA quantifying the effect of strain, sex, and strain × sex interaction on the variability in each surveyed quantitative phenotype.**
(TXT)

**S21 Table. Results from non-parametric Kruskal-Wallis tests assessing the effect of geographic sample location on each surveyed phenotype.**
(TXT)

**S22 Table. Results from one-way ANOVA to quantify the effect of geographic sample location on the variability in each surveyed quantitative phenotype.**
(TXT)

**S23 Table. Results from likelihood ratio tests evaluating the fit of nested linear mixed-effects models to assess the significance of geographic sample location, sex, and location × sex interaction terms on the variance associated with each trait.** Trait values were first regressed on body weight at 18 weeks and residuals used for model fitting.
(TXT)

**S24 Table. Tab-delimited table of all phenotype data.** Table is organized in long-data format, with each row containing the value for a single animal and single phenotype.
(TXT)

**S25 Table. Strain-level summary statistics for each phenotype.** The table is organized in long-data format, with each row containing summary statistics for a single strain and phenotype.
(TXT)

**S26 Table. Spearman correlations for body composition phenotypes over time for each strain and sex.**
(CSV)

**S27 Table. Results from pairwise permutation tests comparing glucose tolerance test response curves between strains.** Male and female curves were analyzed independently owing to pronounced sex differences. Permutation tests were performed using the *CompareGrowthCurves* function in the statmod R package.
(TXT)

**S28 Table. Phenotypic variation within the control strain C57BL/6J across phenotyping cohorts.**
(TXT)

**S29 Table. Statistical comparison of clinical chemistry phenotypes in C57BL/6J mice in the CGDpheno1 dataset on Mouse Phenome Database and C57BL/6J animals in this study.**
(TXT)

**S30 Table. Statistical comparison of clinical chemistry and EKG phenotypes in C57BL/6J mice in the CGDpheno3 dataset on Mouse Phenome Database and C57BL/6J animals in this study.**
(TXT)

**S31 Table. Pairwise strain-level Spearman correlations between surveyed phenotypes.** Correlations were computed using phenotype data from females only.
(TXT)

**S32 Table. Pairwise strain-level Spearman correlations between surveyed phenotypes.** Correlations were computed using phenotype data from males only.
(TXT)

**S33 Table. Tabular synopsis of the terminal tissue collection and tissue processing protocol.**
(XLSX)

## Acknowledgments

We thank Gabriela Heyer, Ketki Samel, David Manahan, Katie Ferris, Henry Thomas, Noelle Bittner, Felipe M. Martins, Sarah Banker, Brett Haines, and Kristin Lapointe for assistance with animal husbandry and breeding. We gratefully acknowledge the contribution of the Genome Technologies, Histology, Research Necropsy, and Clinical Chemistry Scientific Services at The Jackson Laboratory for their contributions to this work. We especially thank members of the Center for Biometric Analysis for their critical contributions toward phenotype data collection.

## Author Contributions

**Conceptualization:** Beth L. Dumont, Gary A. Churchill, Cathleen Lutz, Nadia Rosenthal, Jacqueline K. White, Michael W. Nachman.

**Data curation:** Beth L. Dumont.

**Formal analysis:** Beth L. Dumont, Daniel M. Gatti.

**Funding acquisition:** Beth L. Dumont, Gary A. Churchill, Cathleen Lutz, Nadia Rosenthal, Jacqueline K. White, Michael W. Nachman.

**Investigation:** Beth L. Dumont, Daniel M. Gatti, Lydia K. Wooldridge, Hilda Opoku Frempong.

**Methodology:** Beth L. Dumont, Daniel M. Gatti, Jacqueline K. White, Michael W. Nachman.

**Project administration:** Beth L. Dumont, Gary A. Churchill, Jacqueline K. White, Michael W. Nachman.

**Resources:** Beth L. Dumont, Mallory A. Ballinger, Dana Lin, Megan Phifer-Rixey, Michael J. Sheehan, Taichi A. Suzuki, Cathleen Lutz, Jacqueline K. White, Michael W. Nachman.

**Supervision:** Beth L. Dumont, Gary A. Churchill, Cathleen Lutz, Nadia Rosenthal, Jacqueline K. White, Michael W. Nachman.

**Visualization:** Beth L. Dumont.

**Writing – original draft:** Beth L. Dumont, Michael W. Nachman.

**Writing – review & editing:** Beth L. Dumont, Daniel M. Gatti, Mallory A. Ballinger, Dana Lin, Megan Phifer-Rixey, Michael J. Sheehan, Taichi A. Suzuki, Lydia K. Wooldridge, Hilda Opoku Frempong, Raman Akinyanju Lawal, Gary A. Churchill, Cathleen Lutz, Nadia Rosenthal, Jacqueline K. White, Michael W. Nachman.

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
