## [Decision Letter · Decision Letter 0]

5 Feb 2024

Dear Dr Dumont,

Thank you very much for submitting your Research Article entitled 'Into the Wild: A novel wild-derived inbred strain resource expands the genomic and phenotypic diversity of laboratory mouse models' to PLOS Genetics. Our apologies for the delay in reaching a decision, which was due to difficulty securing external reviewers. We have had a single review in hand for some time; after discussion among editorial board members, we have now decided to proceed.

As you will see, the reviewer appreciated the attention to an important topic but identified some concerns that we ask you address in a revised manuscript. We therefore ask you to modify the manuscript according to the review recommendations. 

Yours sincerely,

Gregory S. Barsh

Editor-in-Chief

PLOS Genetics

Gregory Copenhaver

Editor-in-Chief

PLOS Genetics

Reviewer's Responses to Questions

**Comments to the Authors:**

Reviewer #1: This is a remarkably well written paper introducing the wild derived inbred strain panel (the Nachman panel). The authors have done a tremendous amount of work characterizing the genomic and phenotypic diversity of this panel. An import feature that sets these strains apart from classical inbred mouse strains is the absence of artificially directed selection (selection on fitness will nonetheless occur), which make genetic variation in this panel more resembling human populations, thus presenting better models for human traits. In all aspects surveyed, there is substantial genetic variation, making it an attractive resource, especially considering the deep phenotyping across ages as well as planned phenotyping. I recommend publication as is, but here are a few points for the authors to consider during revision. I would like to emphasize that the paper in its present form is outstanding and deserves to be published.

1. It may be helpful to perform a PCA based on the large collection of phenotypes as a way to separate the lines based on phenotypes and compare the separation pattern with that based on genotypes.

2. When annotating variants using VEP, is the phasing between sites taken into account? For example, a frameshift indel can be shifted back in frame by another frameshift indel. If they are considered separately, they may lead to two loss of function events when the protein is in fact functional.

3. The PCA analysis integrating the Nachman strains and other strains needs to be cautious because the variants are called using different technologies, introducing a batch effect.

4. The value of this resource lies at not only the mouse colonies but also the genotype and phenotype data. How will these be accessible to the public? At this time raw data are deposited and other data are provided as supplemental tables, which aren't ideal. Please note that this is beyond the scope of the paper and the authors probably have a plan already. But I think some forward looking discussion in this paper will go a long way.

5. Can the authors provide some hypothetical examples of how one might use this resource? It was mentioned that crossing these lines to CIs, QTL mapping, etc. But those do not have to utilize these strains.

**Have all data underlying the figures and results presented in the manuscript been provided?**

Reviewer #1: Yes

PLOS authors have the option to publish the peer review history of their article (what does this mean?). If published, this will include your full peer review and any attached files.

Reviewer #1: No

---

## [Decision Letter · Decision Letter 1]

18 Mar 2024

Dear Dr Dumont,

We are pleased to inform you that your manuscript entitled "Into the Wild: A novel wild-derived inbred strain resource expands the genomic and phenotypic diversity of laboratory mouse models" has been editorially accepted for publication in PLOS Genetics. Congratulations!

Yours sincerely,

Gregory S. Barsh

Editor-in-Chief

PLOS Genetics

Gregory Copenhaver

Editor-in-Chief

PLOS Genetics

Comments from the reviewers (if applicable):

Reviewer's Responses to Questions

**Comments to the Authors:**

Reviewer #1: The authors have successfully addressed all my comments. I would like to congratulate them for an outstanding paper.

**Have all data underlying the figures and results presented in the manuscript been provided?**

Reviewer #1: Yes

PLOS authors have the option to publish the peer review history of their article (what does this mean?). If published, this will include your full peer review and any attached files.

Reviewer #1: No

**Data Deposition**

http://datadryad.org/submit?journalID=pgenetics&manu=PGENETICS-D-23-01286R1

**Press Queries**

---

## [Editor Report · Acceptance letter]

3 Apr 2024

PGENETICS-D-23-01286R1 

Into the Wild: A novel wild-derived inbred strain resource expands the genomic and phenotypic diversity of laboratory mouse models 

Dear Dr Dumont, 

We are pleased to inform you that your manuscript entitled "Into the Wild: A novel wild-derived inbred strain resource expands the genomic and phenotypic diversity of laboratory mouse models" has been formally accepted for publication in PLOS Genetics! Your manuscript is now with our production department and you will be notified of the publication date in due course.

With kind regards,

Anita Estes

PLOS Genetics

On behalf of:
